# The long-term impact of BVOC emissions on urban ozone patterns over central Europe: contributions from urban and rural vegetation

Marina Liaskoni[1], Peter Huszár[1], Lukáš Bartík[1], Alvaro Patricio Prieto Perez[1], Jan Karlický[1], and Kateřina Šindelářová[1]

[1]Department of Atmospheric Physics, Faculty of Mathematics and Physics, Charles University, Prague, V Holešovičkách 2, 18000, Prague 8, Czech Republic

**Correspondence:** Peter Huszár (peter.huszar@matfyz.cuni.cz)

**Abstract.** The paper evaluates the long-term (2007-2016) impact of Biogenic Volatile Organic Compounds (BVOC) emissions on urban ozone patterns over central Europe, specifically focusing on the contribution of urban vegetation using a regional climate model offline coupled to chemistry transport model. BVOCs are emitted by terrestrial ecosystems and their impact is considered especially important over NOx-rich environments such as urban areas. The study evaluates the impact of BVOC emissions on ozone ($O_3$), formaldehyde (HCHO) and hydroxyl radical (OH) near surface concentrations, showing an increase in summer ozone by 6-10% over large areas in central Europe due to their emissions. It also demonstrates a substantial increase in formaldehyde concentrations. Additionally, the impact of BVOC emissions on hydroxyl radical concentrations shows a decrease over most of the modelled region by 20-60%, with some increases over urban areas. Impacts on peroxy radicals ($HO_2$ and higher $RO_2$) are shown too.

Importantly, the study explores the partial role of urban vegetation in modulating ozone and evaluates its contribution to the overall ozone formation due to all BVOC emissions. The findings reveal that urban BVOC emissions contribute to around 10% of the total impact on ozone and formaldehyde concentrations in urban areas, indicating their significant but localized influence.

The study also conducts sensitivity analyses to assess the uncertainty arising from the calculation of the urban fraction of BVOC emissions. The results show that the impact of urban BVOC emissions responds to their magnitude nearly linearly, with variations of up to fourfold, emphasizing the importance of accurately quantifying the urban BVOC fluxes. Overall, the study sheds light on the intricate relationship between urban vegetation, BVOC emissions, and their impact on atmospheric chemistry, providing valuable insights into the regional chemistry of BVOC emissions over central Europe and the causes of urban ozone pollution.

## 1 Introduction

Biogenic Volatile Organic Compounds (BVOCs) are atmospheric organic trace gases which are emitted by terrestrial ecosystems. The function of these emissions is connected to the protection of these ecosystems against environmental changes (Simpraga et al., 2019) or herbivory (Yu et al., 2021), as a way of signaling communication, which enhances their growth and their

reproduction. Emission inventories have shown that the prominent species are the terpenoids (i.e. isoprene, monoterpenes, sesquiterpenes), followed by alcohols, carbonyls and acids. Annually global isoprene emissions can reach values of 440 Tg $yr^{-1}$ and contribute to 50% to the total BVOC emissions (Guenther et al., 2012; Sindelarova et al., 2022). Measured BVOC concentrations showed their chemical lifetimes ranging between few minutes to hours, and their reactivity is dependent on many atmospheric factors (Kesselmeier and Staudt, 1999). Due to their high reactivity, BVOCs affect significantly the chemistry of the lower troposphere, by reacting with hydroxyl radical (OH), nitrate radical ($NO_3$) and ozone ($O_3$) leading also to the formation of secondary organic aerosols (Seinfeld and Pandis, 2016; Gu et al., 2021; Bartík et al., 2024).

Several factors influence the magnitude of BVOC emissions. Firstly, the meteorological factors such as the temperature and the solar radiation intensity, both having a positive correlation with the emission rates (Guidolotti et al., 2019). Grote et al. (2013) proved that temperature sensitivity of these emissions depends also on different tree and plant species. Humidity is also another factor that affects the stress response of the plants. Droughts are linked with elevated BVOC emissions, whereas higher levels of humidity can reduce these emissions by closing the stomata of the leaves (Duan et al., 2023). The existence of herbivores and pathogens in the environment adds another factor that contributes to the BVOC emission rates. Injuries from these organisms can stress the plants, resulting in higher emissions (Fitzky et al., 2019).

BVOCs have a complicated role in the tropospheric chemistry. By reaction with OH radical, BVOCs are oxidized to organic peroxy radicals ($RO_2$) following with the reaction with NO to oxidize into $NO_2$ if sufficient NOx is available (e.g. in urban areas). $NO_2$ then undergoes photolysis leading to ozone formation (Coates et al., 2016; Li et al., 2019). However, for low-NOx areas and/or those already rich of VOC, $RO_2$ will react with each other or with $HO_2$ with even a small negative effect on tropospheric $O_3$ (Lerdau, 2007; Seinfeld and Pandis, 2016; Zhao et al., 2022). Apart from contributing to ozone formation, BVOC can also reduce their abundance as $O_3$ is another oxidant leading to formation of Criegee bi-radical intermediates that consequently become aldehydes and ketones (ending in formaldehyde). Finally, during nighttime, the main oxidizing mechanism is the reaction with nitrate radical dominating over the reaction with nighttime ozone (Seinfeld and Pandis, 2016).

It was also shown (e.g. Harrison et al., 2006; Seinfeld and Pandis, 2016) that Criegee intermediates further decompose to hydroxyl radicals and can significantly impact the OH budget over urban areas being a dominant source for OH besides the well-known ozone photolysis pathway.

Among the wide family of BVOCs, a large attention was given to isoprene being the most abundant biogenic species - especially on its oxidation and its impact on recycling of hydroxyl radical and peroxy radicals ($HO_2$ and higher $RO_2$). Archibald et al. (2010) pointed out the complicated pathways isoprene undergoes and showed that any simplification made in the chemical representation (as usually done in chemistry mechanisms within current chemistry transport models) reduces the HOx (OH+$HO_2$) recycling leading to lower OH values compared to measurements. More recently, Bates et al. (2019) argued too that the distribution of its products is highly dependent on the addition of OH and $O_2$ to isoprene and its proper representation allows more prompt regeneration of OH. This consequently improves the ozone-NOx chemistry leading more accurate ozone simulation in numerical models (Schwantes et al., 2020).

In summary, BVOCs affect the HOx and ozone concentrations and thus have a great impact on the oxidative capacity of the lower troposphere which refers to the ability of the atmosphere to remove air pollutants and trace gases (Thompson, 1992).

Moreover, BVOC can oxidize into low volatility substances and contribute thus also to secondary organic aerosol (SOA)
formation (Aksoyoglu et al., 2017; Liu et al., 2021; Huszar et al., 2024; Bartík et al., 2024).

Many studies have shown the overall impact of BVOC emissions on tropospheric ozone concentrations. Over global scale, it was found by many that, as expected, biogenic hydrocarbon emissions increase tropospheric ozone concentrations especially over high-NOx regions while they can lead to even some decreases for the low-NOx case (Williams et al., 2009; Zeng et al., 2008; Rowlinson et al., 2020). Over continental scales, works often focused on highly polluted regions of North America or
Eastern Asia, where BVOC emissions are expected to have a large impact on ozone levels. Sartelet et al. (2012) showed that over North America, average summer ozone concentrations are about 10% larger if BVOC emissions are considered. Zhang et al. (2017) performed a detailed source apportionment using Ozone Source Apportionment Technology (OSAT) and showed a similar contribution of BVOC to regional ozone levels over US, while they also applied a brute-force ("zero-out") method which gave even higher contributions, especially over high-NOx areas. Increasing biogenic emissions of VOC within a warming
climate are also examined and Lam et al. (2011) showed that BVOC emissions in future will become even more important with regards to ozone formation. The above mentioned OSAT technique to attribute the simulated ozone concentrations to different sources was applied by Zhang et al. (2017) and found more than 10% contribution to regional ozone concentrations, mainly above and near urban areas. Sakulyanontvittaya et al. (2016) over Canada showed that region specific landuse and plant type data are needed to achieve better performance in modelling regional scale ozone, especially over urban areas where the effect
of BVOC was found to be higher.

Over highly populated areas in Eastern Asia, ozone is also strongly modulated by BVOC emissions, especially in connection with NOx plumes over downwind areas near large cities (Kim et al., 2013; Lee et al., 2014; Liu et al., 2018; Li et al., 2018; Wu et al., 2020). Elevated role of BVOC in ozone formation during heatwaves when BVOC emissions are higher than average was calculated by Ma et al. (2019). Qu et al. (2013) and Gao et al. (2022b) pointed out that the interplay of anthropogenic
and BVOC emissions is synergical leading to higher ozone concentrations than the sum of their separate contribution. It was shown for eastern Asia that the change of BVOC emissions is responsible for the observed ozone increases (Wang et al., 2022) during the 21 century. The role of BVOC modifications within the changing climate over China was also of interest and due to increase in BVOC emissions ozone increases are also foreseen (Liu et al., 2019).

Over Europe, studies in the Mediterranean region showed that the maximum $O_3$ increase due to the BVOC emissions can be
of order of 10 $\mu gm^{-3}$ (Thunis and Cuvelier, 2000). The dominant role of natural VOC emissions over anthropogenic ones in ozone production (i.e. that ozone concentrations are sensitive mostly to BVOC emissions) over the Mediterranean was shown also by Richards et al. (2013), especially over urban downwind areas, as highlighted earlier by Im et al. (2011). Previously, Curci et al. (2009) calculated for four hot years (1997, 2000, 2001, 2003) an average 5% increase of daily ozone maxima as a result of BVOC emissions, especially during the hottest summers. Many others pointed out the crucial role of BVOC emissions
in air-quality during heatwaves (Castell et al., 2008; Strong et al., 2013; Hodnebrog et al., 2012). Castell et al. (2008) and Tagaris et al. (2014) showed further that the magnitude of the impact of BVOC depend on the NOx emission magnitude and reduction of these emissions in cities would decrease the BVOC impact on ozone. A more elaborated source attribution of European ozone was conducted by Karamchandani et al. (2017) who used OSAT and found too that biogenic emissions are

important for summer ozone. As for the uncertainty of the effect on ozone due to the method of BVOC flux calculation, Jiang et al. (2019) showed that despite the "large differences in isoprene emissions (i.e. 3-fold), the resulting impact in predicted summertime ozone proved to be minor". In central Europe, the produced ozone concentrations due to BVOC emissions are on average 12% higher and can reach values up to 60% on warm days in Berlin (Churkina et al., 2017).

It is clear from the studies above that, from an ozone production perspective, BVOC emissions are important above urban areas with high NOx pollution where there is sufficient NO to be oxidized by peroxy radicals. Certainly, BVOC concentration "clouds" arriving over cities can have large effect on ozone in this regard too (von Schneidemesser et al., 2011), however, urban areas in Europe are never totally non-vegetated, so the VOC emitted by urban vegetation might have a very important role too. These emissions will be probably lower than emissions from rural/natural areas but urban vegetation injects BVOC right into a NOx-rich environment resulting potentially in an efficient ozone production (Gao et al., 2022a; Huszar et al., 2022) and being a dominant factor in the regulation of urban ozone (Fitzky et al., 2023). Moreover, the physiology of trees in urban environments differ from their rural/natural counterparts due to the different urban conditions. E.g. the so-called urban heat island effect creates higher temperatures compared to the rural environments (Huszar et al., 2018; Karlický et al., 2018, 2020). The increased $CO_2$ concentrations can enhance the growth of vegetation and the release of BVOC emissions, a process which is also highly dependent on the plant/tree species (Yu and Blande, 2021). The low levels of soil moisture and humidity act also as a catalyst in increasing emissions. Fitzky et al. (2019) showed that the size of urban trees and the canopy play an important role in mitigating the air pollution, with higher trees being more suitable for this goal. Lastly, the global trend of urban greening sollution poses a further threat to urban ozone as it introduces new BVOC emissions in cities that can potentially enhance ozone concentrations (Ma et al., 2021; Gu et al., 2021)

A few studies already analyzed the partial role of the urban vegetation on air-quality in general and specifically, on ozone formation and removal via deposition. Nowak et al. (2000), showed by analysing various micro-climatic conditions above the urban domain of Washington DC to Massachussets that the uptake of $O_3$ due to BVOC emissions increases more than the formation of $O_3$ during daytime, while in nighttime the formation of $O_3$ dominates the uptake. Lerdau (2007) discussed that in an atmosphere of existing elevated $O_3$ concentrations the physiology of the plants will be affected in a way that will produce further BVOC emissions creating a positive feedback loop of $O_3$ formation. Ghirardo et al. (2016) observed how the urban trees affected the emissions and the uptake of BVOCs. Between constitutive non-stressful conditions and stress-induced BVOCs, the stress-induced constituted about 40% of the total annual BVOC emissions, and this budget could be doubled with the increasing urban greening, highlighting the importance of the urban parameters in BVOC emission models. Calfapietra et al. (2013) and Bonn et al. (2018) further stressed that the choice of the urban trees within urban greening has a crucial importance too with high BVOC emitter trees being dangerous in typical urban conditions (i.e. VOC-limited ones). Using a box-model to analyse the urban micro-climatic conditions, Simon et al. (2019) showed that increasing isoprene concentrations as a result of urban greening can significantly increase the street level ozone concentrations. Recently, Maison et al. (2024) calculated the role of urban vegetation in ozone and organic matter formation for Paris during a chosen summer which included a heat-wave too. They found that ozone increase in average about 2-3% during summer while the heatwave causes even larger, 4-6% increase of near surface ozone.

In summary, there is a generally well established knowledge on the role of BVOC emissions on ozone formation, however, most of the studies looked at only selected months or seasons, or selected urban areas. It is certainly true, that emissions of VOC from vegetation are most important during the hottest days of the year, however the long-term regional impact of BVOC over decadal times-scales has not been well addressed. Moreover, very few studies looked at the partial role of the urban vegetation in modulating ozone, especially in long terms. Here, we try to address these gaps and propose a regional chemistry transport model based study to evaluate the long term impact of BVOC emissions on present day ozone values in and around urban areas over central Europe. Our study moreover calculates the partial role of the urban vegetation and evaluates its contribution to the overall ozone formation due to all BVOC emissions. Apart from ozone, the study further assesses the changes in the oxidative capacity of the atmosphere in terms of OH and $RO_2$ concentrations as well as impact on the products of BVOC oxidation (formaldehyde).

A further motivation of the study is to better understand the causes of urban ozone pollution over Europe. According to the European Environmental Agency Europe's air quality status 2022 (EEA, 2022), 12% of the EU population is exposed to elevated ozone burdens (taking the 120 $\mu gm^{-3}$ as threshold value for the daily maximum 8-hour ozone) while if the WHO guidelines are taken into account (100 $\mu gm^{-3}$) this percentage increases to 95%.

## 2 Methodology

### 2.1 Models used

To achieve the objectives of the study, the chemical transport model CAMx was offline coupled to regional climate model WRF. Biogenic emissions were calculated by the MEGAN model. These three models are described in detail below.

#### 2.1.1 Driving meteorological model

The BVOC emission model used as well as the chemical transport model were driven by the WRF (Weather Research and Forecasting) model version 4 (Skamarock et al., 2019) using the following parameterizations: RRTMG scheme (Iacono et al., 2008) for radiation, Purdue Lin scheme (Chen and Sun, 2002) and the Grell-3D scheme (Grell, 1993) for microphysical processes and convection, respectively, the Noah scheme for the land surface exchange (Chen and Dudhia, 2001) and, finally, the BouLac scheme (Bougeault and Lacarrère, 1989) to resolve the boundary-layer processes. Static land-use data for WRF is derived from CORINE Land Cover data, version CLC 2012 (CORINE, 2012). For urban grid-boxes, the single-layer urban canopy model (SLUCM;(Kusaka et al., 2001)) is used with parameters for the urban built-up same as in Karlický et al. (2018). The choice of the combination of parameterizations follows the results of Karlický et al. (2020) who performed a series of sensitivity experiments to achieve the best possible model-observation agreement.

### 2.1.2 Chemical transport model

To account for the chemical transformation and transport of chemical species, we used the chemical transport model CAMx version 7.20 (Comprehensive Air-quality model with Extensions; Ramboll (2022); Emery et al. (2024)). CAMx is an Eulerian chemical transport model to calculate photochemistry as well as aerosol processes. The CB6r5 gas-phase chemical mechanism (Carbon Bond 6 revision 5) was used in this study. It is described in detail by Cao et al. (2021) with the complete list of chemical species and reactions in Ramboll (2022). Here we provide the most important oxidation pathways for BVOC, namely for isoprene (ISOP) and monoterpenes (TERP).

In CB6r5, ISOP is oxidized most efficiently by reaction with OH radical followed by a lumped specie called ISO2 ("peroxy radicals followed by the reaction of OH with ISOP"). ISO2 then enters reaction chain starting with adding NO forming $NO_2$ along with further products like organic nitrates, formaldehyde, methacrolein, methyl vinyl ketone, including the formation of peroxy radicals ($HO_2$ and higher $RO_2$). ISO2 can also react with other peroxyradicals (including acetyl peroxy radicals). Oxidation of ISOP by ozone is slower in CB6 and included in one summary reaction resulting in formation of formaldehyde (HCHO), methyl vinyl ketone, methacrolein, aldehydes, lumped parafins, hydroxyl radical and peroxy radicals (both $HO_2$ and higher $RO_2$). This means that Crigee biradicals, as intermediate products of this oxidation pathways are not explicitly included in the mechanism. CB6r5 considers steady-state approximation for them and only further products from their decay are considered. Finally (and especially during night), ISOP is oxidized in CB6r5 by the nitrate radical ($NO_3$) resulting in formation of HCHO, higher aldehydes, methyl vinyl ketone, $NO_2$ and organic nitrates as well as peroxy radicals. The reaction of ISOP with nitrate is faster than with ozone, but still slower than the OH oxidation. The main ISOP oxidation products (methyl vinyl ketone, methacrolein) are further oxidized again by either OH, ozone or $NO_3$ ending in simpler aldehydes, ketones, methyl glyoxal, formic acid, hydroxyl radical and of course some peroxy radicals.

In case of monoterpenes, they are represented with one lumped specie (called TERP) that represent all monoterpenes. Their oxidation in CB6r5 follows again three pathways (i.e. reaction with OH, ozone and nitrate radical). The fastest oxidation occurs with OH followed by the reaction with nitrate radical and ozone. These oxidation pathways are again represented by three summary reactions with the products involving HCHO and higher aldehydes, organic nitrates, parafins, hydroxyl radical and peroxy radicals. Again, Criegee intermediates are not explicitly calculated, only the consequent products (which include also hydroxyl radical). The monoterpene oxidation pathways are based on the older CB05 gas-phase chemistry mechanism (Sarwar et al., 2008) which uses the reaction rate constants taken from the SAPRC99 mechanism (Carter, 2000). These reaction constant represent an average of the individual reaction constants for different compounds that comprise the monoterpene family.

It is seen that the oxidation of ISOP and TERP in CAMx is represented by a relatively simple set of reaction equations and does not account for the intermediate reaction steps and products including possible feedbacks including the consideration of Crigee intermediates. In this regards, the Master Chemical Mechanism (MCM; https://mcm.york.ac.uk/MCM/about/, last access 25 SEPT 2024) could provide a much more detailed degradation scheme or there are attempts to incorporate special schemes aimed at only isoprene (Bates et al., 2019) or both isoprene and terpenes (Schwantes et al., 2020) that better represent e.g. the HOx recycling and thus the impact on NOx-ozone chemistry as detailed by Archibald et al. (2010) with recent updates

provided by Khan et al. (2021). CAMx is however intended to be employed over large domains and in case of longer numerical integrations like in this study some compromise has to be made between the complexity of the chemistry and the computational feasibility. On the other hand, the "summary" reactions for TERP and ISOP mentioned above may introduce some error in the distribution of the oxidants and effort should be made towards more explicit mechanism in future model works (e.g. the new CB7 mechanism; Schwantes et al., 2020).

As gas-phase chemical reactions are tightly coupled to heterogenous chemistry and aerosol processes, we also invoked the full aerosol chemistry in our simulations using the ISORROPIA thermodynamic equilibrium model v1.7 (Nenes et al., 1998, 1999) and RADM-AQ aqueous chemistry algorithm. A semi-volatile equilibrium scheme called SOAP (Strader et al., 1999) is used to form secondary organic aerosol from condensable vapours.

CAMx is offline coupled to WRF output using the wrfcamx preprocessor that is provided with the CAMx code https://www.camx.com/download/support-software/ (last access: 25 SEPT 2024). Vertical eddy-diffusion coefficients (Kv) are computed in wrfcamx following the similarity method adopted from the CMAQ model (Byun, 1999). The sensitivity to the choice of the method for the calculation of Kv was tested by Huszar et al. (2020a) and they found that the CMAQ method provides Kv values in the middle of the uncertainty range.

### 2.1.3 The biogenic emission model

Biogenic emissions from terrestrial ecosystems are calculated offline with the MEGANv2.1 (Model of Emissions of Gases and Aerosols from Nature) model (Guenther et al., 2012). MEGAN provides meteorology dependent emission fluxes of a whole range of biogenic volatile compounds such as isoprene, monoterpenes and sesquiterpenes as well as methanol, ethanol, acetaldehyde, acetone, $\alpha$-pinene, $\beta$-pinene, t-$\beta$-ocimene, limonene, ethene, and propene etc. MEGAN2.1 estimates emissions ($F_i$) of chemical species $i$ from terrestrial landscapes according to:

$$F_i = \gamma_i \sum_{j=1}^{n} \epsilon_{i,j} \chi_j \tag{1}$$

where $\epsilon_{i,j}$ is the emission factor at standard conditions for vegetation type $j$ with fractional grid box areal coverage $\chi_j$. The emission activity factor ($\gamma_i$) accounts for the processes controlling emission responses to environmental and phenological conditions. This includes a light response based on electron transport, a temperature response due to enzymatic activity, and further a dependence on leaf age, soil moisture, leaf area index (LAI) and $CO_2$ concentration (called $CO_2$ inhibition). MEGAN considers 16 vegetation types called plant-functional types (PFT) and emission factors for 19 biogenic species that are speciated to the target chemical mechanism (CB6r5 in our case). The meteorological input to MEGAN is obtained from WRF hourly outputs. The output of MEGAN are hourly emission fluxes for the target grid (see further).

### 2.2 Model setup and data

The WRF model as well as CAMx were run on a central European domain of size 189 $\times$ 165 gridboxes (from France to Ukraine, Italy to Denmark) at 9 km $\times$ 9 km horizontal resolution centered over Prague (Czechia) (50.075N, 14.44E; Lambert

Conic Conformal projection). WRF used 40 layers in vertical reaching 50 hPa as model top while the lowermost layer was about 30 m thick. CAMx was applied over 18 layers with the top one at about 12 km. The first ten CAMx layers matched the WRF layers.

Simulation were carried out for the 2007-2016 period. This length (10 years) ensures higher representivness of the results providing "climatology" of the air-quality impact of BVOC emissions.

To drive the regional climate in WRF, the ERA-interim reanalysis (Simmons et al., 2010) was used. Chemical initial and boundary conditions for CAMx where adopted from CAM-Chem global model data (Buchholz et al., 2019; Emmons et al., 2020).

The TNO-MACC-III data (an update of the MACC-II version; Kuenen et al. (2014)) from 2011 were used as anthropogenic emission data for the whole decade. This high resolution (1/8° longitude 1/16° latitude, roughly 6 km x 6 km) European emission database provides annual emission totals for non-methane volatile organic compounds (NMVOC), NOx, methane ($CH_4$), sulfur dioxide ($SO_2$) , ammonia ($NH_3$), carbon monoxide (CO) and PM10 and PM2.5 (particles with diameter less than 2.5 and 10 microns, respectively) in 11 activity categories. To redistribute the emissions to model grid-cells, the FUME

version 2.0 (Flexible Universal Processor for Modeling Emissions) emission model was used (Benešová et al. (2018); Belda et al. (2024), http://fume-ep.org/, last access: 25 SEPT 2024). FUME further performs the standard emission processing steps, like chemical speciation and time disaggregation to hourly emissions with speciation profiles and temporal factors based on Passant (2002) and van der Gon et al. (2011). The output of the FUME are CAMx-ready hourly emission files speciated to model species (consistent to the used mechanism).

For MEGAN the MODIS $0.1° \times 0.1°$ LAI data from year 2010 at 8 day period was used (Yuan et al., 2011). Plant-functional-type data are extracted also from MODIS following Lawrence and Chase (2007) for the same year (2010). Finally, emission factors for different plant types are based on Guenther et al. (2012). To justify to choice of one year as a representative one for the whole period, we plot the average monthly timeseries of LAI from the Yuan et al. (2011) year-by-year data as domain average in Fig. 3. The plot shows that for each simulated year, the LAI for summer months reaches slightly above 2.5 in average

(about 2.6-2.7) while the 2010 year is not an outlier meaning that it is a good representative for the decade (there are larger differences during winter between the years but we were interested only in the summer months in this study). Regarding the plant-functional-type data, here we assume negligible modifications in land-use over central Europe during this decade. This is true especially for crops and other farm land types and also for forests (EUROSTAT, 2020; FORESTEUROPE, 2020), at least compared to areas where intense urban development as China (Zhu et al., 2022) took place in the last decades requiring to take

the evolution of PFT distribution and LAI much more into account (Ma et al., 2021).

MEGAN was driven with WRF-generated hourly meteorological output resulting in hourly BVOC emissions files on a daily basis for the entire 2007-2016 period.

### 2.2.1   The modelled BVOC emissions

Fig. 1 depicts the 2007-2016 average summer (June-July-August, JJA) emissions of BVOC as well as the sum of all anthro-

pogenic VOC emissions for the same period. This allows to compare the magnitudes of both sources.

As one of the objectives of the study is to evaluate the partial role of BVOC emissions originating from urban areas, the figure shows also the urban BVOC emissions for the six selected cities. The urban fraction was calculated by masking out the PFT data by the city boundaries in a way that firstly, it was calculated that how much of the gridcell falls within the urban area and then this factor was applied to the PFT fractional data allowing to calculate the emissions of BVOC from this fraction only.

From Fig. 1 it is seen that BVOC emissions are usually between 10 and 40 $molkm^{-2}hr^{-1}$ with emissions up to 60-80 $molkm^{-2}hr^{-1}$ over natural areas over the southern part of the domain (Italy and the Balkans). In case of anthropogenic VOCs (expect methane), they are usually lower over non-urban areas ranging up to 5 $molkm^{-2}hr^{-1}$ being thus smaller than the biogenic source. However, over cities - as expected - the anthropogenic source of VOC is much stronger, reaching 100-150 $molkm^{-2}hr^{-1}$. The domain wide average of anthropogenic and biogenic emissions of non-methane VOCs is 5.2 vs. 18.9 $molkm^{-2}hr^{-1}$, respectively. This means that VOCs have a roughly 3x larger biogenic source over the area compared to anthropogenic ones during the summers in the examined decade, however, they have of course substantially different spatial distribution.

The BVOC emissions originating from cities was calculated by masking out the total biogenic emissions fields with the individual cities administrative boundaries based on shapefiles provided by GADM public database (https://gadm.org, last access 08 May 2024). As the resolution of the MEGAN inputs (LAI and PFT) as well as the output resolution were approximately the same and relatively coarse (around 0.1 degree which is about 10 km in Central Europe) , this was the only possible method for an estimate of the urban-portion of the BVOC emissions. However, we admit that the urban BVOC emissions can be higher or lower depending on the exact distribution of the vegetation in and around cities. Therefor this study also contains sensitivity experiments aiming to evaluate the uncertainty of the results to the magnitude of the urban portion of the BVOC emissions (see further). The resulting BVOC emissions from urban vegetation is depicted in the right panel of the figures and shows that these emission are usually below 20 $molkm^{-2}hr^{-1}$ however, for some cities there are model gridboxes with emissions up to 60 $molkm^{-2}hr^{-1}$. On the other hand, gridboxes that match the inner part of cities can exhibit almost zero BVOC emissions (like in case of Berlin, Prague or Budapest) creating a ring-like emission pattern.

As the modelled BVOC fluxes are dependent on the meteorology supplied and this meteorology is marked with some biases, we also compared the modelled emissions fluxes with emissions that were driven with more accurate meteorological data. For this purpose, the CAMS-GLOB-BIO3.1 data from the Copernicus Atmosphere Monitoring Service (Sindelarova et al., 2022) was chosen. These data are provided at $0.25°$ x $0.25°$ resolution so somewhat comparable to our resolution and they are driven by the ERA5 reanalysis. The results are depicted in Fig. 2. Both emissions data (those from MEGAN and from CAMS) lie between 0 and 40 $molkm^{-2}hr^{-1}$, but the CAMS fluxes are usually smaller. Over central Europe CAMS fluxes reach often only 0.5-2 $molkm^{-2}hr^{-1}$ while at the same locations our emissions are around 5-10 $molkm^{-2}hr^{-1}$. On the other hand, over southern Europe, the isoprene emission peaks are similar in magnitude and have similar geographic distribution (southern France, Italy, Balkans). The reason for these differences is probably in the fact that the 3.1 version of these data incorporated updated region-specific emission factors for different plants instead of using the default MEGAN emission factors (used in version CAMS-GLOB-BIO2.1 as well as in our study) and this resulted in lower emissions over Europe compared to the default one used in our set-up (see the difference between version 2.1 and 3.1 in the mentioned study). Moreover, CAMS-

GLOB-BIO3.1 was calculated based on ERA5 while the earlier version CAMS-GLOB-BIO2.1 as well as WRF were driven by ERA-Interim. Karlický et al. (2018) who used WRF driven by ERA-Interim over the same domain (and resolution) and with a similar set of parameterizations showed some overestimation of near surface temperatures in central Europe which is connected to the overall positive bias in ERA-Interim temperatures. Consequently, we can expect that the temperatures in our simulations are also higher that they would have been using ERA5. Lower temperatures in ERA5 thus also add to the difference between our BVOC fluxes and the ones in CAMS-GLOB-BIO3.1.

In summary, these results indicate that our emissions fluxes might be somewhat overestimated over the central part of the domain which probably means that the effects on ozone are overestimated as well.

## 2.3 Model simulations

Multiple model experiments (simulations) were made by CAMx depending on whether all BVOC emissions or only the urban/nourban fraction of them were accounted for. The list of simulations is presented by Tab. 1. Note, that each of these CAMx simulations were driven by the same WRF simulation, i.e. no differences between the driving meteorology between the individual simulations were present (the same WRF outputs were used to drive the CAMx model also in Liaskoni et al. (2023); Karlický et al. (2024)). In the first simulation denoted "allBVOC" all BVOC emissions were considered, i.e. those from rural and natural areas as well as those originating from urban vegetation. In the "noBVOC" simulation, BVOC emissions were removed from the entire domain. For partitioning between the effect of the urban portion of BVOC emissions, we performed a further simulation were all BVOC emissions were considered except those originating from the six selected cities (called "nuBVOC" as only "nourban" BVOC are considered) and a simulation were only the urban portion of the BVOC emissions is considered ("uBVOC"). To assess the sensitivity of the results to the amount of urban BVOC emissions, i.e. the way how the urban and rural fraction of BVOC emission is calculated, we performed two additional simulations, "2nuBVOC" and "0.5nuBVOC" were the fraction of the BVOC emissions falling within the boundaries of the urban area is 2x larger and half of the original fraction, respectively. These sensitivity simulations were carried out only over a three year long period from 2007 to 2009.

## 3 Results

### 3.1 Model validation

A comparison with both rural and urban AirBase station measurements for ozone and its precursor $NO_2$ has been performed in order to assess the CAMx's ability to capture the long term monthly and daily variability of these pollutants. For ozone, 117 rural stations were used in total while for each of the six selected city, we chose all urban and suburban background stations that lie within the city (resulting in 2 to 5 stations per city). For $NO_2$ we used stations from the previous selection that contain also this pollutant (to maintain consistency between the validation results for ozone and nitrogen dioxide). The model validation over rural stations aims to show the overall model performance in describing regional scale concentrations, while the urban

comparison serves to evaluate the model strength and weaknesses over individual urban areas which logically have a different ozone regime (towards VOC-limited compared to the rural NOx-limited one). The complete list of the stations used is provided in the Supplement.

In Fig. 4 the annual cycle of the monthly means and the summer average diurnal variation for rural stations is shown while we averaged over all stations (the corresponding standard deviation is shown too). The annual cycle is overestimated during almost all months with the best match in winter. In summer, model values are overestimated by up to 15 $\mu gm^{-3}$. The summer diurnal cycle shows that while the daily maxima are overestimated only slightly (by 5 $\mu gm^{-3}$), the nighttime values are higher in CAMx by almost 20 $\mu gm^{-3}$. This means that the overestimated nighttime ozone is probably the main cause of the summer overestimation of monthly means.

Over urban areas (Fig. 5) the summer overestimation is even larger reaching 20 $\mu gm^{-3}$ while the nighttime ozone values are higher in model by more than 30-40 $\mu gm^{-3}$. Moreover, over cities, the ozone daily maxima are also overestimated (by around 5-10 $\mu gm^{-3}$) while the maxima are reached often a bit earlier than in the measured values.

The annual cycle of $NO_2$ monthly means (Fig. 6a) shows a systematic underestimation by 2-3 $\mu gm^{-3}$ while the largest occurs during January to April (up to 4 $\mu gm^{-3}$). The overall shape of the annual cycle is however captured very well. The measured summer diurnal cycle (Fig. 6b) shows two maxima during morning and evening hours corresponding to morning and evening rush hours. These are somewhat captured in CAMx too but the concentrations are underestimated while the largest negative bias occurs between 8 and 12 a.m. local time (2-3 $\mu gm^{-3}$).

The underestimation of nitrogen dioxide is even larger over urban areas, but the magnitude is different across different cities. As monthly averages (Fig. 7; left) the negative bias ranges from 5 $\mu gm^{-3}$ (for Vienna) to some 10-15 $\mu gm^{-3}$ seen over Budapest while it is usually larger during the cold season. The summer diurnal cycles show that the model has a tendency to predict the two measured $NO_2$ maxima however their modelled timing is often not well captured (occurs a bit earlier) or the evening maximum is not present at all (over Berlin, Budapest or Warsaw). Also the underestimation is large and often exceeds 30 $\mu gm^{-3}$ (but is usually around 10 $\mu gm^{-3}$).

## 3.2 The impact of all BVOC emissions

The spatial and temporal distribution of the impact of all BVOC emissions on ozone, formaldehyde and hydroxyl radical is presented in this section as the difference between the "allBVOC" and "noBVOC" experiments (see Tab. 1). In case of ozone and formaldehyde, the relative plots denote the relative contribution of BVOC emissions to the total concentrations calculated as (allBVOC-noBVOC)/allBVOC $\times$ 100%. For OH the relative changes are calculated after introducing BVOC emissions, i.e. as (allBVOC-noBVOC)/noBVOC $\times$ 100%.

The impact on 2007-2016 summer (JJA) average MDA8 ozone is presented in the upper row of Fig. 8 and it shows that ozone increases by 2-4 ppbv (6-10% relative contribution) over large areas in central Europe with maxima over urbanized areas exceeding 6 ppbv (10-12%) while the highest impact is modelled over northern Italy exceeding 12 ppbv (15-20%).

We also plot, in the same figure (Fig. 8), the main oxidation product of BVOC, formaldehyde to see how it is increased due to this natural VOC source. The average JJA HCHO increases usually by 0.5-1 ppbv which is around 40-60% in relative

contribution showing a substantial increase due to biogenic emissions. Over urban areas however the relative contribution is smaller due to the large anthropogenic VOC source over cities. The highest HCHO increases are modelled over the southern part of the domain aligning with the largest BVOC emissions reaching 1-2 ppvb or 60-80% in relative numbers.

Finally, as the hydroxyl radical is a key oxidant that is modulated by the presence of VOC it is desirable also to plot the impact of BVOC emissions on OH concentrations. Here, we are interested in the daytime OH values, so the impact on the JJA average daily maxima is plotted. Due to BVOC emissions, OH decreases over most of the domain by around 0.1-0.5 $\mathrm{pgm}^{-3}$ (20-40%) while the decrease over urban areas is smaller or even some increases are modelled over cities. Over areas without BVOC emissions (sea) however OH increases by around 0.01-0.02 $\mathrm{pgm}^{-3}$ (2-5%).

Apart from the average impacts for summer, we also plot the day-by-day evolution of the impact on MDA8 ozone for six selected urban areas over central Europe (Berlin, Budapest, Munich, Prague, Vienna and Warsaw) in Fig. 9. We also included the results for the vicinity of these cities to evaluate the difference in impact over urban centres with high-NOx environment vs. urban surroundings, where the NOx/VOC ratio is much lower leading to potentially smaller BVOC impact. The plot shows that, as expected, the winter impacts are small, barely exceeding 1 ppbv. The JJA impacts are on the other hand usually above 5 ppbv but extreme values in selected years can exceed 15 ppbv (especially for Berlin and Vienna, where it is exceeded at least once each year). It is also seen that the impact over urban surroundings is somewhat smaller with maximum impacts lower by 2-3 ppbv compared to the maximum impacts over city centres.

Additionally, in the supplement, we present the spatial impacts calculate also for individual simulated year. This also helps to reveal the inter-annual variability of the impacts (Figures S1-S6).

We also plot the diurnal cycle of the BVOC impact on the three analysed species to analyse the variation of BVOC role during different parts of the day. Fig. 10 presents the results for the six cities and their vicinity along with the absolute ozone concentrations. The diurnal variation of ozone values has the expected shape with minima around sunrise and maxima during early afternoon with values in cities between 25-30 ppbv and 45-50 ppbv for the minimum and maximum values, respectively. Over city surroundings, the corresponding values are slightly lower during afternoon and evening hours (by around 5 ppbv). The impact of BVOC has a well described shape during the day with minimum values aligned with the minima of the absolute ones (around 1.5-2.5 ppbv). During morning hours, there is a rapid increase of the impact reaching maxima around noon (up to 4-5 ppbv) exhibiting a slower decrease during afternoon and evening time. There is a indication for a small secondary maxima (for some cities) during late afternoon which might be caused by elevated emissions of NOx during this part of the day (afternoon rush hour). The impact of BVOC over city surrounding has a slightly different shape with minima occurring around the same time as over city centres with similar values. However, the maxima are lower by around 1 ppbv.

Analogically to $O_3$, Fig. 11 depicts the diurnal cycles for formaldehyde. The absolute HCHO concentrations show a rather uniform pattern during the day with values around 1.5-2.5 ppbv for all city centres while over city surroundings the values are usually lower during the daytime while for nighttime, they can be higher than city centre values for some cities (Budapest, Prague and partly Munich too) but are usually lower owing to smaller VOC emissions compared to city centres that decompose into formaldehyde. The impact of BVOC emissions has a double-peak pattern with highest values during nighttime reaching 0.7-1.2 ppbv while lowest values occur usually during early afternoon (0.6-1 ppbv). A secondary maximum of the BVOC

impact is visible in Budapest, Munich, Prague and (to a smaller extent) Berlin during morning hours reaching 0.6-1.2 ppbv. Over urban vicinity, the impact of BVOC on HCHO is slightly lower during the day compared to the impact over centres but is about 0.1-0.2 ppbv higher during evening hours.

Finally, the diurnal variation of the absolute urban OH concentration and the impacts of BVOC emissions is plotted in Fig. 12. The absolute OH concentrations have the well known diurnal pattern with very low residual values during night and a strong peak during noon when solar insolation is at maximum. The maxima over urban centres reach about 0.25-0.35 $\mathrm{pgm}^{-3}$ while the maxima over surrounding is somewhat lower around 0.2 $\mathrm{pgm}^{-3}$. The impact of BVOC emissions is substantially different between the city centres and their vicinity. While over vicinity, there is a strong decrease during the day peaking at around -0.1 to -0.2 $\mathrm{pgm}^{-3}$, over urban centres, first there is a slight increase during morning hours turning to decrease during noon and to an increase again during evening hours. However, in some cities (Vienna and Warsaw) the decrease of the impact during noon does not lead to negative values.

## 3.3   The impact of urban BVOC emissions

The partial impact of BVOC emissions from vegetation within the selected urban areas is plotted in Fig. 13. The impact is calculated as the difference of "allBVOC" and "nuBVOC" simulations and the domain is "zoomed" to the six cities (i.e. not showing the entire domain). Alternatively, we calculated the impact of urban BVOCs also as the difference of simulations "uBVOC" and "noBVOC" in Fig. 14. Thus, the impact is calculated with respect to two reference states, the first one considers all BVOC emissions except those in cities, while the second one considers no BVOC emissions at all (at least within the computational domain).

The impact of urban BVOC on ozone is a clear increase up to 0.6 ppbv over city centres while a few 10 km away from cities the impact reduces below 0.1 ppbv. HCHO increases due to urban BVOC by about 0.08 ppbv, with maximum values often over the edges of the cities which have already large vegetation cover and still belong to the city outskirts. This "ring"-like feature (seen also on the emission plots) is seen also for OH, which decreases due to urban BVOC by around 0.1-0.2 $\mathrm{pgm}^{-3}$. If the impact of urban BVOC is calculated from the "noBVOC" background reference, it is clearly larger. The $O_3$ increases reach 0.8 ppbv and the diameter of the 0.1 ppbv increase around cities is larger. Larger impact is seen also for HCHO, where the increase reaches or even exceeds 0.1 ppbv. Finally, OH also decrease in larger extent reaching -0.2 $\mathrm{pgm}^{-3}$. This clearly shows the importance of the choice of reference state towards which the impact is calculated.

In a similar fashion, as for the impact of all BVOC emissions, we plot also the diurnal cycle of the impact of urban BVOC in Fig. 15-17 for $O_3$, HCHO and OH radical, respectively. We also plot the absolute values for the "allBVOC" and "nuBVOC" case.

The impact of urban BVOC on ozone over city centres has a distinct diurnal cycle with minimum values over night up to 0.02 ppbv while the maxima occur around noon reaching 0.1 to 0.4 ppbv depending on the city (maximum for Vienna). Unlike the impact of all BVOC, this impact has only one peak which is much narrower than in the former case. The impact over vicinity is (as expected) much smaller, reaching around 0.05 ppbv with the peak occurring a few hours later than the peak over centres.

For formaldehyde, the urban BVOC impact has a very similar shape to ozone with maximum impact in city centres occurring around noon reaching 0.08 to 0.13 ppbv (highest for Prague) with nighttime impact often below 0.01 ppbv. Over city vicinity, the impact remains very small peaking around 0.01-0.02 ppbv during noon.

    Finally, the impact of urban BVOC on OH over urban centres has a distinct negative peak occurring during noon. It usually reaches -0.01 to -0.02 $pgm^{-3}$ with maximum decreases over Munich and Prague reaching -0.04 $pgm^{-3}$. The impact over

vicinity is almost negligible with a small peak around noon too.

    The quantification of the magnitude of the partial impact of urban BVOC allows us to calculate their relative contribution to the impact of all BVOCs. This was calculated for ozone and formaldehyde and plotted in Fig. 18. OH was omitted from this figure as the impact of all BVOC is positive while the impact of the urban vegetation is negative.

    For ozone, urban BVOC has a contribution to the total impact over city centres up to 8-10% (highest share over Vienna and

Berlin). In case of HCHO, the contribution has a similar magnitude peaking at 8-10%. For OH, we cannot clearly define its "contribution" to the total OH changes, however, comparing the positive impact of all BVOC to the negative impact of urban BVOC, these are very similar in absolute numbers. This indicates that if one considers only the urban vegetation, the impact can be qualitatively totally different from the impact of considering all vegetation (i.e. also from those above rural areas).

### 3.4   Impact on hydroxyl and peroxide radicals ($OH, HO_2, RO_2$)

Given their importance in urban (but also rural) tropospheric chemistry and atmospheric oxidation of VOCs including ozone formation, and to facilitate the interpretation of the modelled impacts on ozone and formaldehyde, we further present the impact of BVOC emissions on peroxide radicals ($HO_2$ and higher $RO_2$). For completeness we also included the impact on average OH (besides the impact on daily maximum OH presented above) to account for the whole HOx family.

    Fig. 19 shows that the impact on absolute OH is about -0.02 to -0.08 $pgm^{-3}$ over large regions with largest decreases over

the southern parts of the domain reaching -0.1 $pgm^{-3}$. This means that the decrease of average values is about one third of the decrease of daily maximum ones. Over urban areas (similar to daily maximum values) concentrations tend to increase due to BVOC by up to 0.02-0.04 $pgm^{-3}$. In relative numbers, OH decreases by about 20-60% reaching 80% over southern parts of the domain. Over cities, the increase is about 10%.

    Hydroperoxyl radical shows a clear increase due to BVOC emissions above 1 $pgm^{-3}$ over many areas being highest over the

southern part of the domain reaching 4-5 $pgm^{-3}$. In relative numbers this means a 40-80% increase over rural areas, however, over highly polluted regions (mostly urban areas) the increase is over 100% reaching as much as 300%.

    Finally for total peroxy radicals ($RO_2$) we modelled a 10-40 $pgm^{-3}$ increase over rural areas (especially over the southern part of the domain) while the increase over urban areas is smaller, usually below 10-15 $pgm^{-3}$. In relative numbers, the $RO_2$ increase over rural areas due to BVOC emissions is remarkable, reaching 600-900% while over urban areas, this is about

200-300% confirming that BVOC is a dominant source for $RO_2$.

    Focusing on urban BVOC emissions only, in Fig. 20 we see that the average summer OH is reduced by up to 0.02 $pgm^{-3}$ representing an about -10 to -20% decrease. In case of $HO_2$, it increases due to urban BVOC by up to about 0.3-0.6 $pgm^{-3}$

with peaks up to 1 pgm$^{-3}$ making an about 10-20% increase. For the total peroxy radical concentrations, the increase is about 0.5-1 pgm$^{-3}$ (up to 4 pgm$^{-3}$) over the selected cities meaning an about 10-30% increase (up to 40-50% for Berlin.)

### 3.5 Sensitivity to urban fraction of BVOC emissions

As already mentioned the partition between urban and rural BVOC emissions was based on calculating the fraction of the gridcell that lie within the given city boundaries. This however brings some degree of uncertainty to results as the distribution of vegetation within the given gridcell is usually not uniform . Here we assess this uncertainty by conducting further experiments with reducing/increasing the fraction of BVOC emissions that lie within or outside of urban area. As already mentioned in the Methodology, we took 1) a case where only half of the urban fraction of gridcell area is considered and a case 2) where the urban fraction of gridcell area is 2x that large (upper-bounded by the gridcell total area itself). Of course, if a gridcell is entirely within the urban boundaries, 100% of the BVOC emissions are considered as urban. Fig. 21 shows how the modified urban BVOC emissions differ from the default case in relative numbers. As expected, emissions are either reduced by around 50% with smaller reduction near city edges and larger one near centres while in the second case, the emissions are almost twice as large as in the default case.

Fig. 22 shows under reduced urban fraction of BVOC emissions that impact on MDA8 ozone over city centres reduces to around 0.1-0.2 ppbv while if the opposite is considered, the impact reaches 0.8 ppbv. In other words, it is roughly half or twice as large than in the default case showing strong sensitivity of ozone production on the amount of urban BVOC. For HCHO (Fig. 23), the situation is similar. While at reduced urban BVOC the impact is around 0.02-0.04 ppbv over city centres, in the doubled urban BVOC case the impact reaches 0.08-0.1 ppbv. These numbers represent, again, half and twice of those in the default case, respectively. For the impact on OH radical (Fig. 24) reduced urban BVOC emission results in a negative impact smaller than 0.02 pgm$^{-3}$ (usually even smaller than 0.01 pgm$^{-3}$) in absolute numbers while for increased urban fraction of these emissions the impact is much higher, reaching -0.2 pgm$^{-3}$, more than double of the default impact of urban BVOCs.

## 4 Discussion and conclusions

The study evaluated the present day impact of emissions of biogenic volatile organic compounds on near surface concentrations of ozone and formaldehyde, as well as on the oxidative capacity of the atmosphere in terms of hydroxyl and peroxy radical concentrations.

The comparison with surface measurements performed for ozone showed some distinct patterns in model biases that include: i) overestimation of spring-to-autumn average ozone values, ii) large overestimation of summer night-time ozone concentrations and a smaller overestimation of summer daily ozone maxima, especially over cities. Regarding the first bias pattern (an overestimation over rural areas up to 20 µgm$^{-3}$ while up to 30 µgm$^{-3}$ over cities), similar positive model bias was encountered earlier by Huszar et al. (2020a) for the same region for both urban and rural stations, who found that it occurs at all used horizontal resolutions (3/9/27 km). Later, Huszar et al. (2020b) confirmed this bias too. It is probably caused mainly by the large night-time overestimation of ozone while the daytime values are captured more accurately. This behavior was seen in

many previous similarly oriented studies (e.g. Karlicky et al., 2017; Huszar et al., 2018; Otero et al., 2018) or recently in de la Paz et al. (2024) and is caused probably by inaccurate vertical mixing in the nocturnal boundary layer as pointed out by Zanis et al. (2011), although the nocturnal ozone chemistry (CB6 in our case) is marked with deficiencies too (Im et al., 2015). It is thus clear that more emphasis should be made to improve the modelling of night-time ozone to get a more reasonable starting point for the daytime ozone formation (Wong and Stutz, 2010). Another feature seen is the higher daily ozone maxima

during summer in model compared to observations, especially over cities. Urban centres are affected by high rate of titration due to concentrated NOx emissions. When the coarse resolution is used as in this study, then these concentrated city center emissions are not resolved and they are instead diluted to the model grid so the increase of NOx concentration is not high enough for the first order ozone titration; instead, it efficiently causes ozone production (Markakis et al., 2015). This behavior was recently seen in Zhu et al. (2024) where the daily urban ozone maxima were often overestimated for the mentioned reason.

The conclusions above are also well supported by the $NO_2$ model biases encountered. In our simulations and especially over cities, nitrogen dioxide is strongly underestimated which is probably connected to the instant dilution of emissions into the model gridbox (of 9 km x 9 km size) as well as overestimated vertical eddy diffusion which removes too much NOx from the lowermost model layer (Huszar et al., 2020a). However, this means ozone removal by titration (as stated above).

We modelled on average an about 6-8 ppbv (up to 12 ppbv over southern Europe) increase of near surface ozone due to all

BVOC emissions. This is in line with previous chemistry transport model studies applied over Europe (Thunis and Cuvelier, 2000; Curci et al., 2009; Richards et al., 2013). E.g. Hodnebrog et al. (2012) found peak impact of BVOC on ozone up to 10 ppbv over the Mediterranean which corresponds well to our results. Moreover, we found that the highest impacts are often over regions with high NOx pollution (Po Valley, western Germany, southern Poland etc.), which is also reasonable as these areas are VOC-limited i.e. any addition of VOC (of any origin, natural or anthropogenic) results in efficient ozone formation (Li et

al., 2018; Gao et al., 2022a).

We have also seen that due to BVOC emissions, formaldehyde concentrations significantly increased (contributing by up to 40-60% ) with smaller increases in cities. This is inline with the fact that HCHO is the main product of the oxidation of most of the hydrocarbons including those of biogenic origin (Luecken et al., 2012; Kaiser et al., 2015; Chen et al., 2023). The strong ties between BVOC emissions and HCHO burdens was studied earlier by many, often to infer biogenic emissions from

HCHO columns (Dufour et al., 2009; Curci et al., 2010) and our results confirm the expectation. Indeed, highest increases in formaldehyde are modelled for southern part of the domain that exhibits the largest emissions. On the other hand, over urban areas with limited biogenic emissions source, the contribution is much smaller. Previously Bastien et al. (2019) modelled the HCHO contributors and found a relative contribution from biogenic source of the order 20-40% which is a bit smaller than our number, 40-60%. This later number also corresponds to the fact that (as domain average) BVOC emissions are about 3

times higher during summer compared to anthropogenic ones, while it has to be kept in mind that not all VOC are oxidized to formaldehyde with the same efficiency. Therefor we cannot assume that the relative contribution to HCHO of a particular VOC source category (biogenic in this case) will equal to the relative contribution of the source to the total VOC emissions.

For the hydroxyl radical, large areas exhibited a decrease of daily maxima while over cities and oceans, increases were modelled due to BVOC emissions. As OH is one of the main oxidants of BVOC (Kelly et al., 2018) (besides the nitrate radical

and ozone), it is clear that introducing BVOC adds new sink to OH making OH daytime concentrations lower while not all OH is recycled. This apparently outweighs the additional OH production from ozone (via the atomic oxygen $O^1D$ and water vapor) or from the ozonolysis of BVOC via Crigee intermediates decay (Seinfeld and Pandis, 2016). On the other hand, this later process is probably the reason for some increases of OH over urban areas. Over them, the BVOC emissions are very small while the ozone increases were large causing OH increases due to direct production from atomic oxygen. Moreover, as

already mentioned in the introduction, the ozonolysis of BVOC (as well as in general of alkenes) leads to Crigee intermediate production which further decomposes yielding additional OH. Averaged over day and night, this may exceed the OH yields from ozone photolysis (Johnson and Marston, 2008).

As for the diurnal variation of the above presented impact in and around cities, it has a very distinct pattern for each three chemical. For ozone, the impact follows more or less the absolute values as a result of the photochemical activity being the

strongest during noon. However, the maximum impact occurs sooner than noon indicating that maybe the maximum of the $NO_2$ production from reaction of NO and peroxy radicals is playing role here. This, besides $RO_2$, is influenced by the available NO, which gets lower towards noon due to decrease of emissions after the morning peak. There is also an indication for a early evening peak for ozone formation due to BVOC which again might be connected to increased NOx emissions during evening rush hours. Over urban vicinities, the peak impact is smaller which is a natural result of lower NOx emissions limiting the $NO_2$

formation due to reaction with peroxy radicals (Seinfeld and Pandis, 2016).

The impact on formaldehyde over urban centres has a more complicated diurnal cycle with often two minima occurring during morning and late afternoon hours, and maximum values during evening. To explain this behavior, we have to understand the sinks and the sources of this chemical. The production of HCHO during daytime is mainly by oxidation of VOC due to OH which is largest when both OH and VOC peak (e.g. Wu et al., 2023; Geo et al., 2023). This would imply a maximum impact

of BVOC during daytime. However, BVOC can be oxidized also by ozone and nitrate radical, which is the main mechanism during nighttime and low solar insolation (and also low OH), so HCHO is produced during night-time too. There is however a strong sink for HCHO during daytime in both reaction with OH and photolysis which is even more important during the day than the reaction with OH (Possanzini et al., 2002; Seinfeld and Pandis, 2016). During night-time the additional HCHO produced from BVOC has thus a suppressed sink leading to higher impact over the night than during the day. The impact

over vicinities is even higher which is a clear consequence of higher BVOC emissions but the suppressed reactions forming peroxy-acetyl nitrates (PAN) due to smaller NOx concentrations can play a role too.

In case of OH, the diurnal pattern of the BVOC impact over city vicinities shows a clear decrease during noon which is in line with the expectation that OH is consumed by the oxidation of BVOC. However, over city centres the pattern indicates that two competing process take place: the first is the already mentioned OH decrease due to oxidation of BVOC. However, the

impact on OH first shows some increase before this decrease occurs. Here probably the fact that extra OH is produced from the BVOC induced ozone via atomic oxygen ($O^1D$) as well as from ozonolysis of BVOC and depending on the counteract between this production and the loss due to BVOC oxidation determines whether the average impact on OH will be negative or positive.

We showed that hydroperoxide radical is increased due to BVOC by more than 200% and this increase is larger over cities. This is due to the fact that absolute $HO_2$ concentrations are much smaller over cities or over areas with high NOx source due to reaction with NO. This in turn caused large relative importance of BVOC emission over such areas in terms of $HO_2$ production. On the other hand, the total $RO_2$ produced is highest in relative (but also absolute numbers) over rural areas which is probably connected to the simple fact that $RO_2$ is largely composed of peroxides originating from BVOC oxidation yielding high relative impact over areas with high BVOC emissions. Or in other words, most of the $RO_2$ mass comes from BVOC oxidation. In both cases ($HO_2$ or the total peroxides) however, the increases clearly explain the impact on ozone, as more NO can be oxidized to $NO_2$ by reaction with peroxy radicals.

Our analysis showed that the urban BVOC emissions alone act rather locally and cause an increase of urban ozone by less than 1 ppbv (usually a few 0.1 ppbv). One of the very important results of this study is that the urban fraction of BVOC emissions contributed to the total impact of all BVOC by around 10%. In case of HCHO, the urban BVOC induced changes are of order less then 0.1 ppbv making this again an about 10% contribution to the total HCHO increases due to all BVOC emissions.

For OH, urban BVOC caused a clear decrease which is a crucial difference compared to the impact of all BVOC. As already detailed above within the diurnal variation of the impacts, multiple competitive processes act to modulate OH modifications due to BVOC emissions. Evidently, if the isolated effect of urban BVOC on OH is analysed, the $VOC + OH$ oxidation dominates over the extra OH produced due to photolysis of increased $O_3$ or from BVOC ozonolysis (via the decay of Criegee intermediates). Also the diurnal cycle of the impact of urban BVOC on OH shows only a clear decrease during the day. The impact of urban BVOC on OH is also very limited to the city outskirts which means that the local BVOC emissions are important mainly for short term chemical effects and radicals with short lifetime (Seinfeld and Pandis, 2016) while the effect on chemicals with longer lifetime, like ozone and formaldehyde, the impact propagates over larger area (a few 10 km from the city centre) as seen in Fig. 13.

We also showed that if the impact of urban BVOC is calculated from a "clean" reference with no BVOC emissions at all, the impact is larger. This can be attributed to the fact that the OH radical competes for less BVOC molecules when urban BVOC are emitted into a "BVOC-free" air making their oxidation more efficient. This consequently leads to more $RO_2$ formed hence more NO oxidized into $NO_2$ leading to enhanced ozone formation. With more BVOC oxidized, more OH is removed leading to larger decreases of OH.

A shortcoming of the study is the way the urban fraction of BVOC is calculated, i.e. based on the fraction of emitting gridboxes that lie within the urban boundaries. To address the uncertainty arising from this approach, we calculated the impact of urban BVOC under an decreased and increased urban fraction corresponding to about 50% and 200% of the original BVOC emissions. The uncertainty analysis showed that the impact on ozone ranges between about 0.2 to 0.8 ppbv, so almost 4 fold difference which well corresponds to the 4 times larger BVOC emissions in the former case. The situation was similar in case of HCHO and OH too. This means that the impact of the urban fraction of BVOC emissions responds to their magnitude nearly linearly. The accurate quantification of the urban BVOC fluxes is therefor crucial and requires first a precise description of the vegetation within the urban built-up (Calfapietra et al., 2013).

In summary, our study showed that ineligible portion of the overall impact of BVOC emissions over central Europe's urban areas is attributable to local BVOC emissions. This is especially true for influencing the oxidative capacity of urban air via OH radical. However, it also stressed a relatively high uncertainty to the quantification of this fraction. Future research should focus on high resolution model assessment of the impact of local BVOC emissions reflecting the large spatial diversity of the vegetation within the urban areas. We also have to add that in overall the BVOC emissions fluxes are marked with uncertainty too (not only their "urban" fraction) as was seen from the comparison with available global biogenic emission data. This uncertainty has to be assessed in future research too.

*Code and data availability.* CAMx version 7.20 is available at https://www.camx.com/download/source/ (last access: 14 May 2024; CAMx, 2022; Ramboll, 2022). WRF version 4.0 can be downloaded from https://www2.mmm.ucar.edu/wrf/src/WRFV4.0.TAR.gz (last access 14 May 2024; WRF (2018)). The MEGAN v2.10 code can be obtained from https://bai.ess.uci.edu/megan/data-and-code/megan21 (last access: 15 May 2024) while the FUME emission model used to be found under https://doi.org/10.5281/zenodo.10142912 (last access: 15 May 2024). The raw CAMx model outputs from all simulations comprise about 20TB of 3-dimensional data of the concentrations of main pollutants and are stored on the authors storage facilities. These are to be obtained upon request. The extracted hourly near surface concentrations (which are the basis of the presented analysis) are available on the Czech National Repository from https://doi.org/10.48700/datst.b6as3-v2y27 (last access 2024 OCT 2) (Huszar, 2024). The observational data from the AirBase database can be obtained from https://discomap.eea.europa.eu/map/fme/AirQualityExport.htm (last access: 15 May 2024) (EEA, 2023).

*Author contributions.* ML and PH conceptualized and designed the experiments and wrote the majority of the text, PH conducted the CAMx simulations, JK performed the WRF experiments, ML, LB, APPP contributed to the analysis of the results and KS helped with processing the biogenic emissions and writing the text.

*Competing interests.* No competing interests are present.

*Acknowledgements.* This work has been supported by the Czech Technological Agency (TACR) grant No.SS02030031 ARAMIS (Air Quality Research Assessment and Monitoring Integrated System), the Charles University Grant Agency (GAUK) project no. 298822, and partly by the Project OP JAK "Natural and anthropogenic georisks" CZ.02.01.014/0022_008/0004605 and Project of EC Horizon no. 101056783 "NON-CO2 FORCERS AND THEIR CLIMATE, WEATHER, AIR QUALITY AND HEALTH IMPACTS" (FOCI). It was partly supported also by the HPC infrastructure of Ministry of Education, Youth and Sports of the Czech Republic through the e-INFRA CZ (ID:90254). We also further acknowledge the TNO-MACC-III emissions dataset provided by the Copernicus Monitoring Service, the compiled air quality station data provided by the European Environmental Agency and the ERA-Interim reanalysis provided by the European Centre for Medium-Range Weather Forecast.

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

**Table 1.** The list of CAMx simulations performed with the information BVOC emissions considered, i.e. their rural (nourban) and urban fraction.

| | Regional Chemistry Transport Model (CAMx) simulations | | | |
|---|---|---|---|---|
| | Experiment | rural BVOC emissions | urban BVOC emissions | period |
| 1 | allBVOC | yes | yes | 2007-2016 |
| 2 | noBVOC | no | no | 2007-2016 |
| 3 | nuBVOC | yes | no | 2007-2016 |
| 4 | uBVOC | no | yes | 2007-2016 |
| 5 | 2nuBVOC | yes (elevated urban fraction) | no | 2007-2009 |
| 6 | 0.5nuBVOC | yes (reduced urban fraction) | no | 2007-2009 |

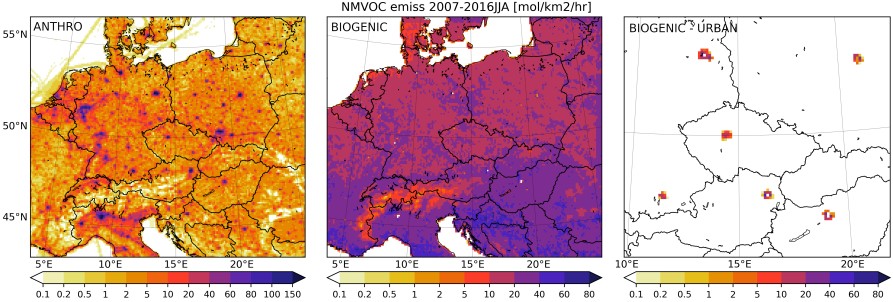

**Figure 1.** The 2007-2016 JJA average emissions of anthropogenic (left) and biogenic (middle) VOC emissions and the urban fraction of BVOC (right) in $\mathrm{mol\,km^{-2}\,hr^{-1}}$. Please, note that the colorbars for the BVOC emission panels are slightly different from the anthropogenic emission figure.

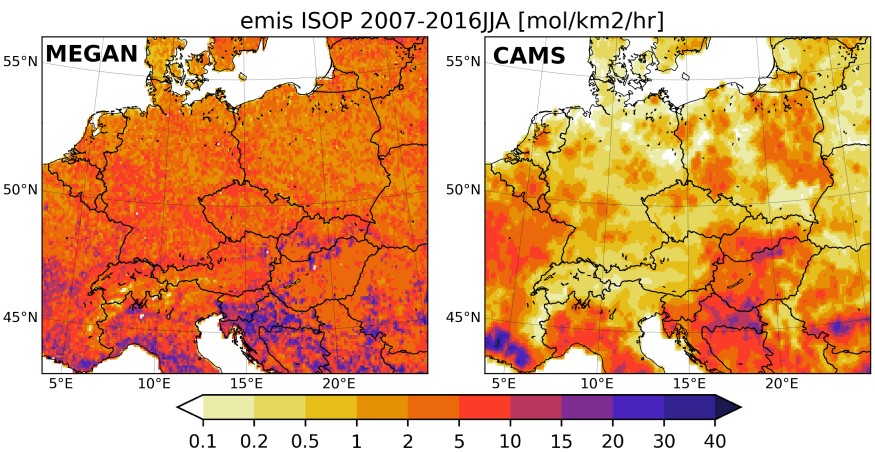

**Figure 2.** Comparison of the 2007-2016 JJA average emissions of Isoprene from biogenic sources from the MEGAN model (left) and from the CAMS global biogenic emission data (CAMS-GLOB-BIO; right) in $\mathrm{mol\,km^{-2}\,hr^{-1}}$.

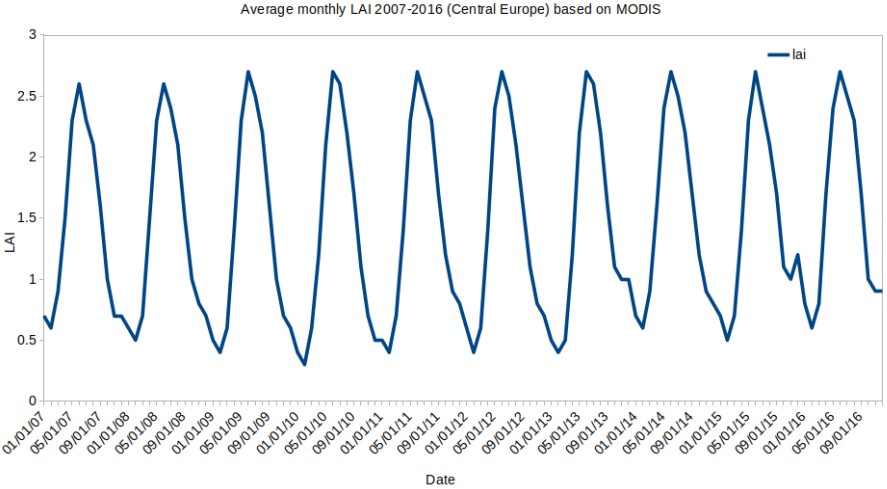

**Figure 3.** Timeseries of the monthly mean leaf-area-index averaged over the domaiun for the 2007-2016 period based on MODIS data.

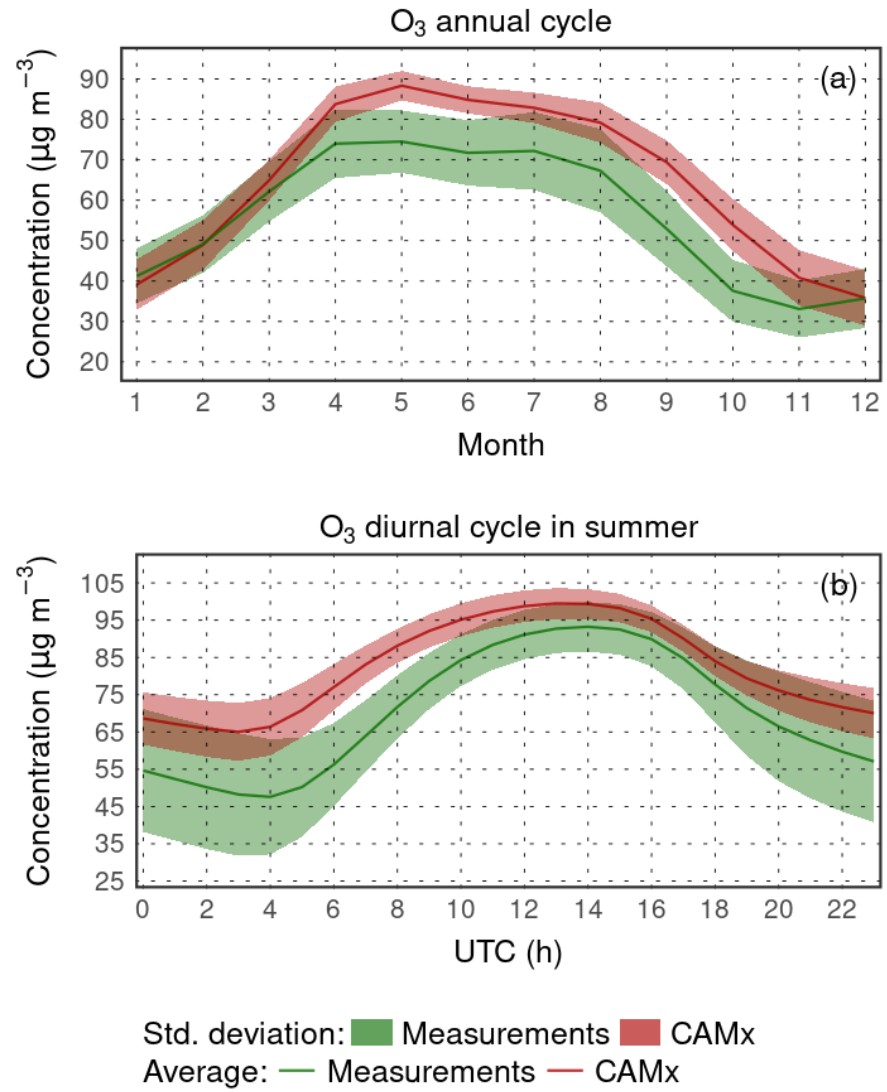

**Figure 4.** Comparison of modelled surface ozone concentrations with rural background stations for the annual cycle of the average monthly means (a) and average JJA diurnal cycle (b) in μgm$^{-3}$.

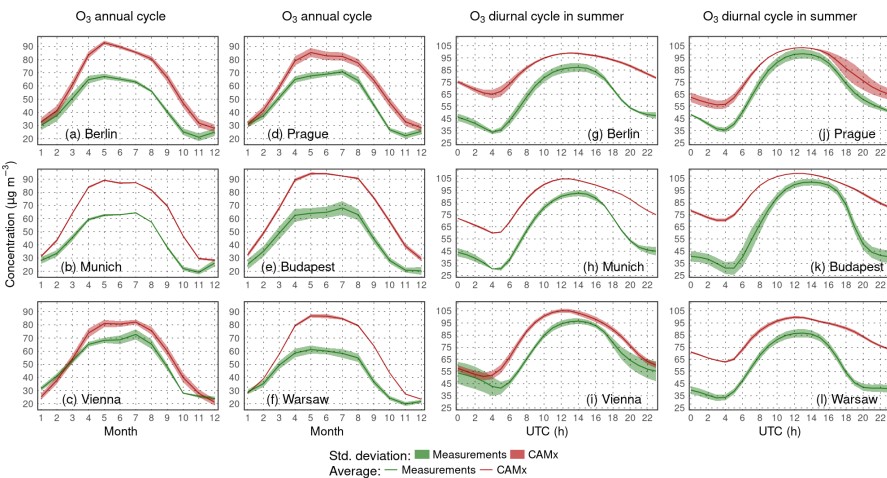

**Figure 5.** Comparison of modelled surface ozone concentrations with urban background stations from six selected city for the annual cycle of the average monthly means (a-f) and average JJA diurnal cycle (g-l) in $\mu g m^{-3}$.

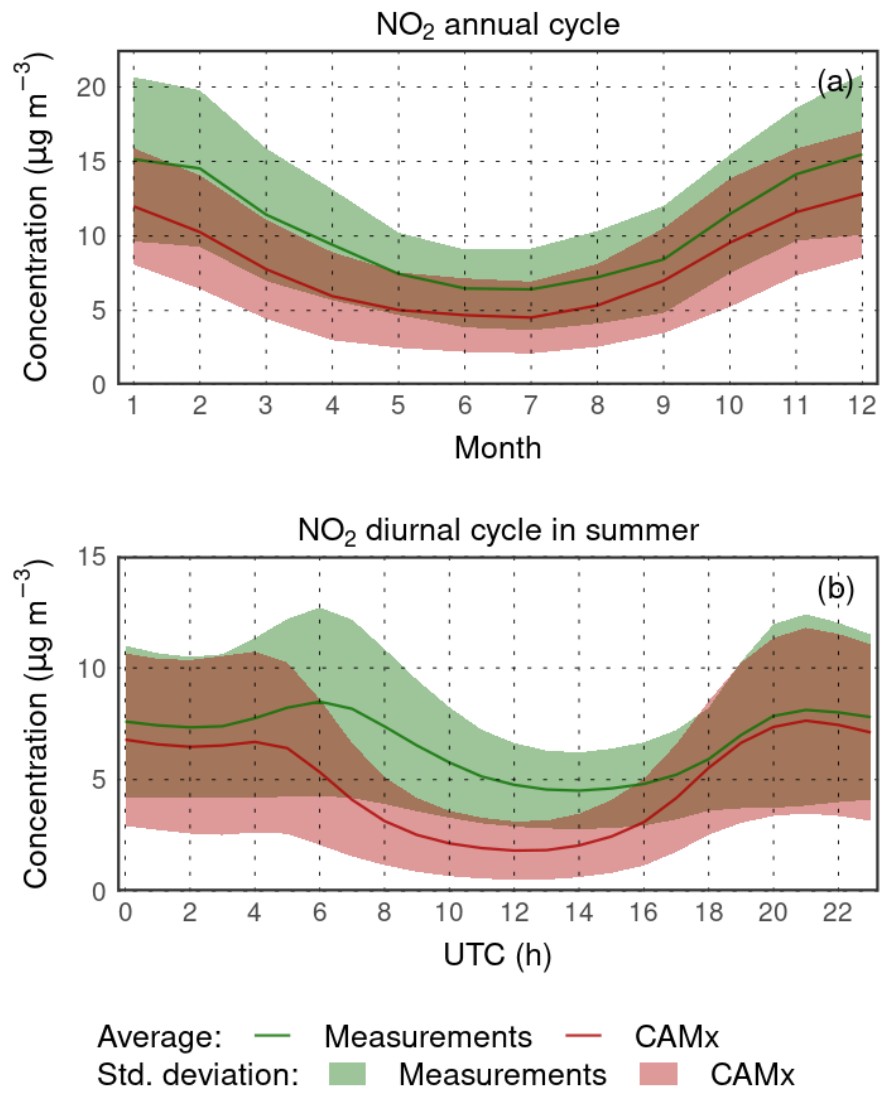

**Figure 6.** Comparison of modelled surface $NO_2$ concentrations with rural background stations for the annual cycle of the average monthly means (a) and average JJA diurnal cycle (b) in $\mu gm^{-3}$.

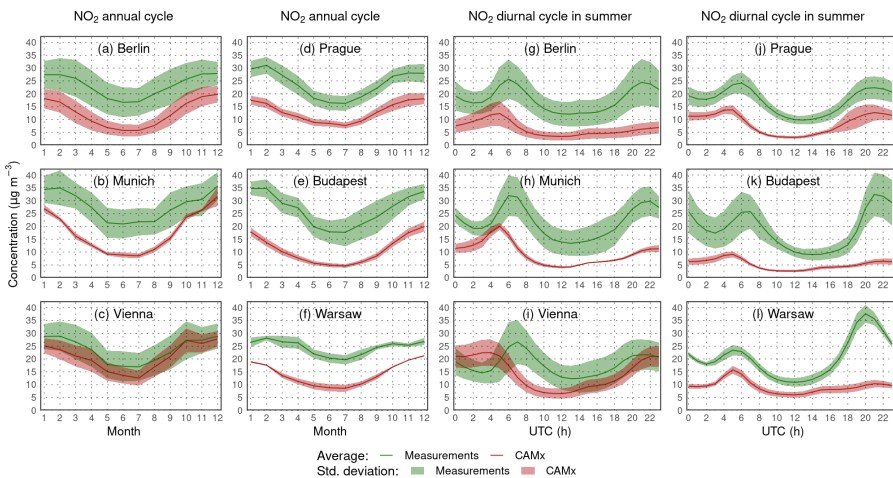

**Figure 7.** Comparison of modelled surface $NO_2$ concentrations with urban background stations from six selected city for the annual cycle of the average monthly means (a-f) and average JJA diurnal cycle (g-l) in $\mu gm^{-3}$.

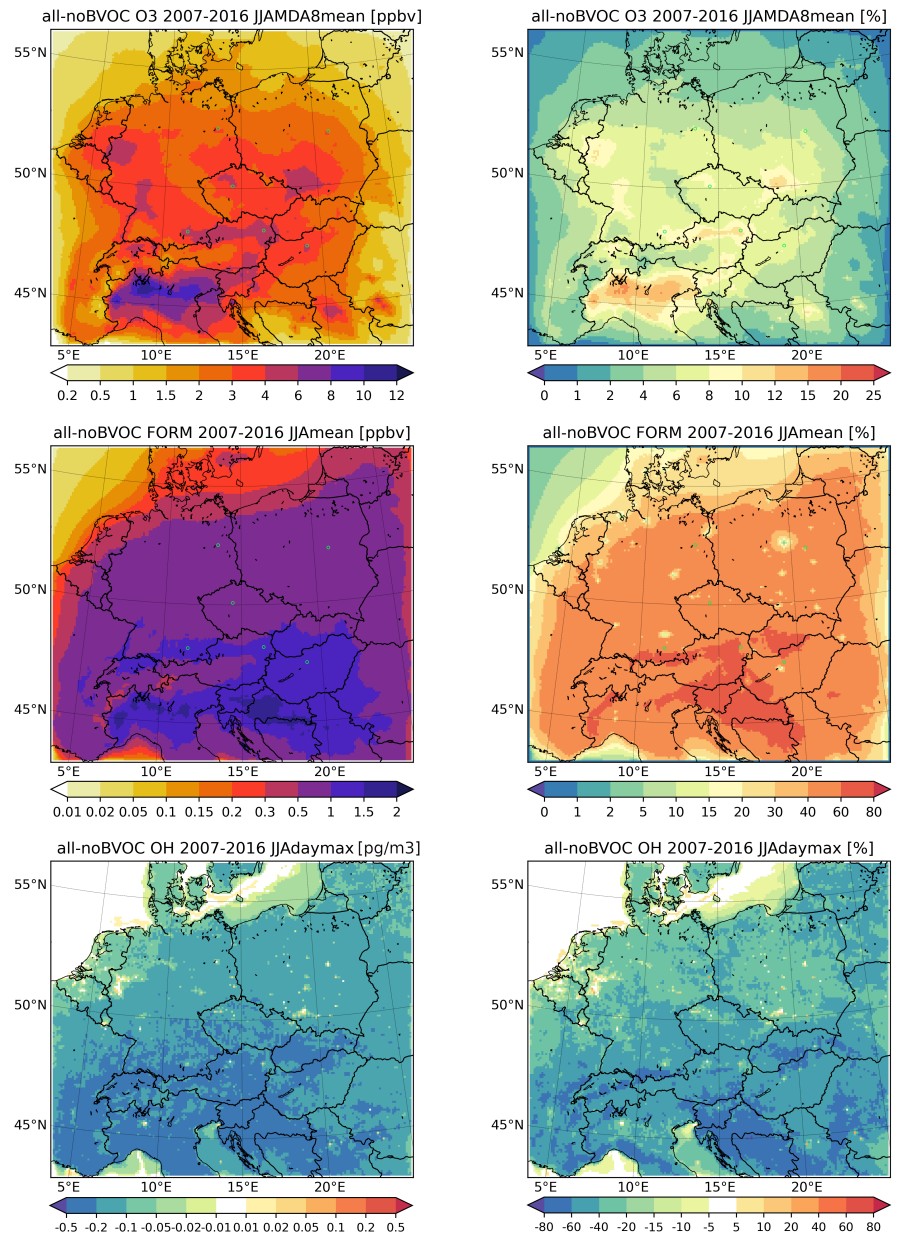

**Figure 8.** The average 2007-2016 JJA impact of all BVOC emissions on the maximum daily 8hour ozone (MDA8; upper row), daily mean formaldehyde and daily max hydroxyl radical near surface concentrations in ppbv for $O_3$ and HCHO (FORM in figure titles) and in $\mathrm{pgm}^{-3}$ ($10^{-6}\mu gm^{-3}$) for OH. Left columns shows the absolute impact, the right shows the relative contribution in %

.

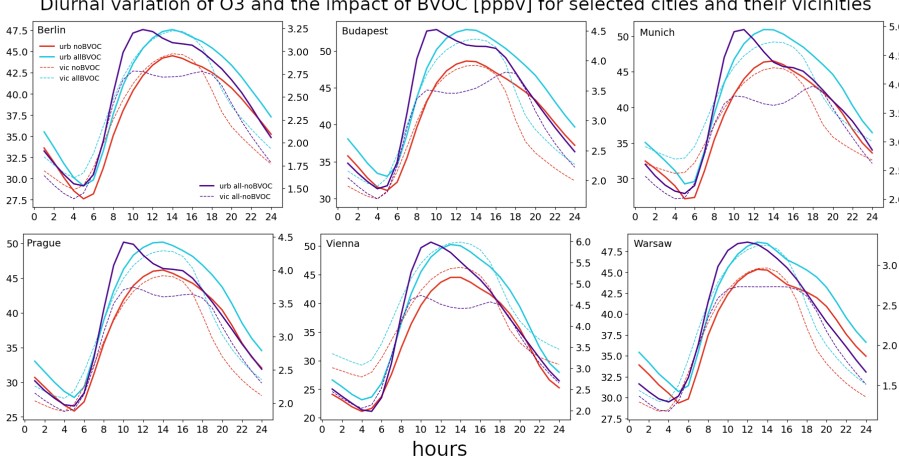

**Figure 9.** The 2007-2016 timeseries of the BVOC impact of MDA8 ozone for the centres and vicinities of 6 selected cities in ppbv

.

**Figure 10.** The average 2007-2016 JJA diurnal cycle of near surface $O_3$ for 6 city centres (bold) and their vicinitites (dashed) for the "noBVOC" (red) and "allBVOC" case (cyan). The right y-axis corresponds to the impact of BVOC emissions (allBVOC-noBVOC; violet). Cities are Berlin, Budapest, Munich, Prague, Vienna and Warsaw. Units in ppbv.

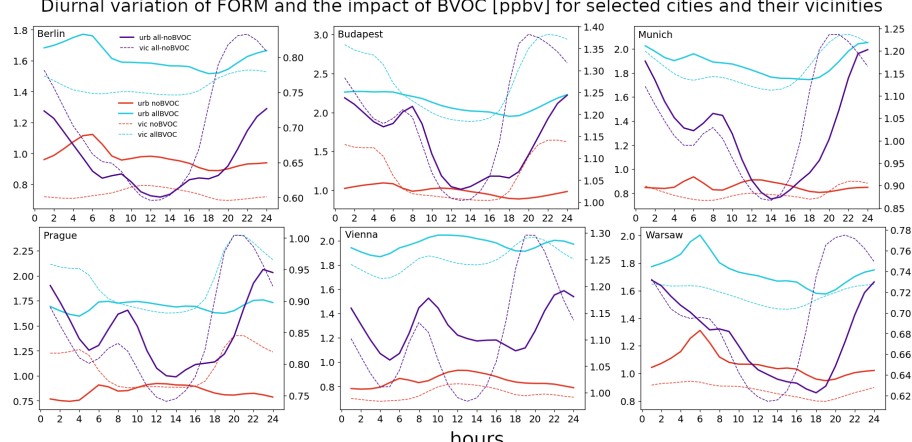

**Figure 11.** Same as Fig. 10 but for formaldehyde.

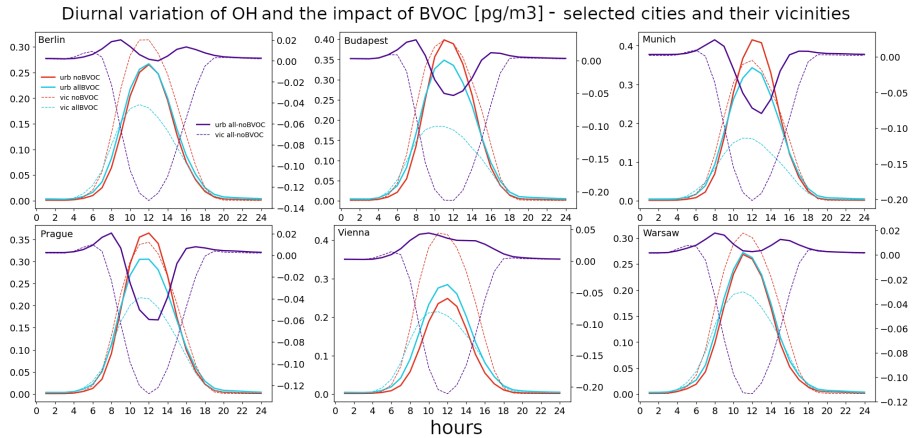

**Figure 12.** Same as Fig. 10 but for OH and in $\mathrm{pgm}^{-3}$ ($10^{-6} \mu gm^{-3}$).

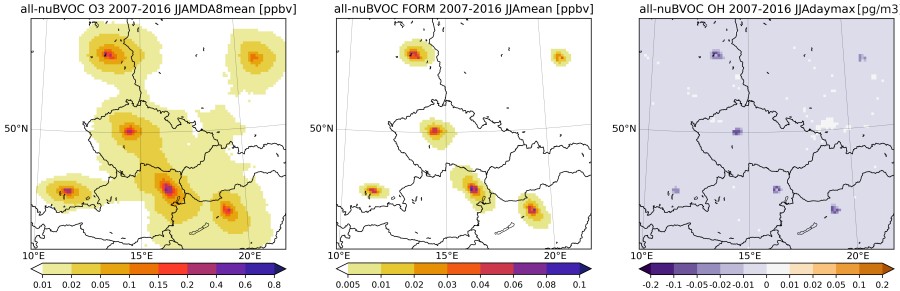

**Figure 13.** The impact of urban BVOC emissions (calculated as allBVOC-nuBVOC) on the MDA8 ozone (left), daily mean HCHO (middle) and maximum daily OH (right) averaged over 2007-2016 JJA. The plot is zoomed to the cities analyzed. Units are ppbv for ozone and FORM, and $\mathrm{pgm}^{-3}$ ($10^{-6} \mu gm^{-3}$) for OH.

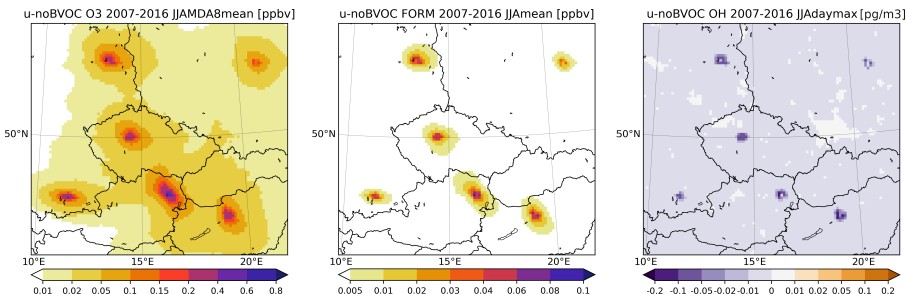

**Figure 14.** The impact of urban BVOC emissions (calculated alternatively as uBVOC-noBVOC) on the MDA8 ozone (left), daily mean FORM (middle) and maximum daily OH (right) averaged over 2007-2016 JJA. The plot is zoomed to the cities analyzed. Units are ppbv for ozone and FORM, and pgm$^{-3}$ ($10^{-6}\mu gm^{-3}$) for OH.

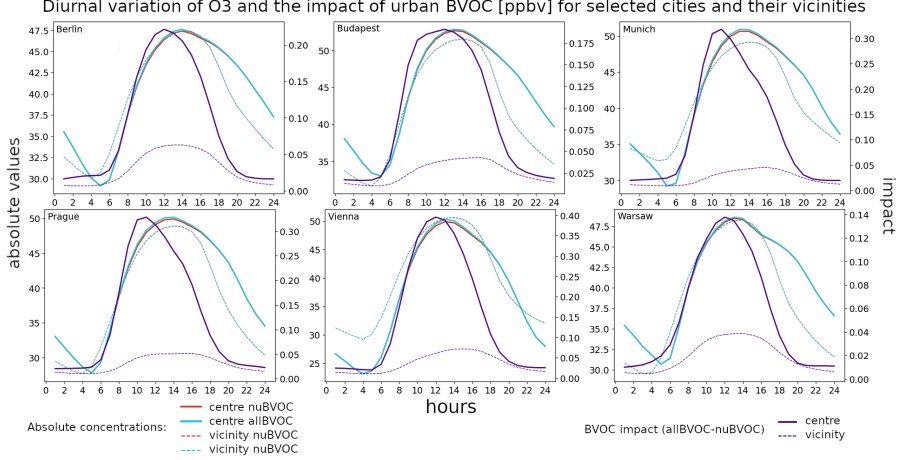

**Figure 15.** The average 2007-2016 JJA diurnal cycle of near surface O$_3$ for 6 city centres (bold) and their vicinities (dashed) for the "nuBVOC" (red) and "allBVOC" case (cyan). The right y-axis corresponds to the impact of urban BVOC emissions (allBVOC-nuBVOC; violet). Cities are Berlin, Budapest, Munich, Prague, Vienna and Warsaw. Units in ppbv.

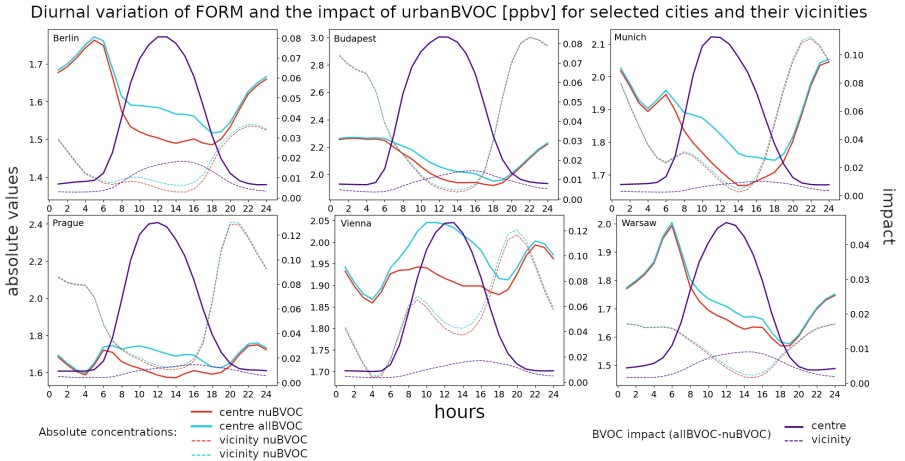

**Figure 16.** Same as Fig. 15 but for FORM.

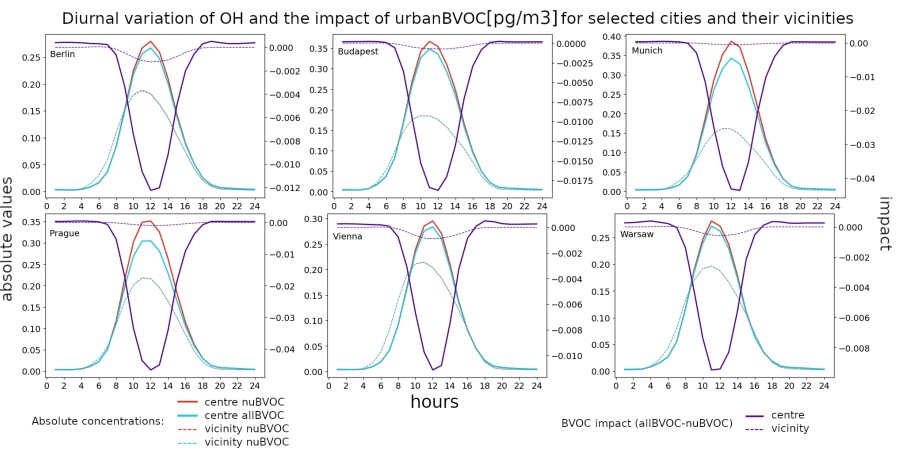

**Figure 17.** Same as Fig. 15 but for OH and in $\text{pgm}^{-3}$ ($10^{-6}\mu gm^{-3}$).

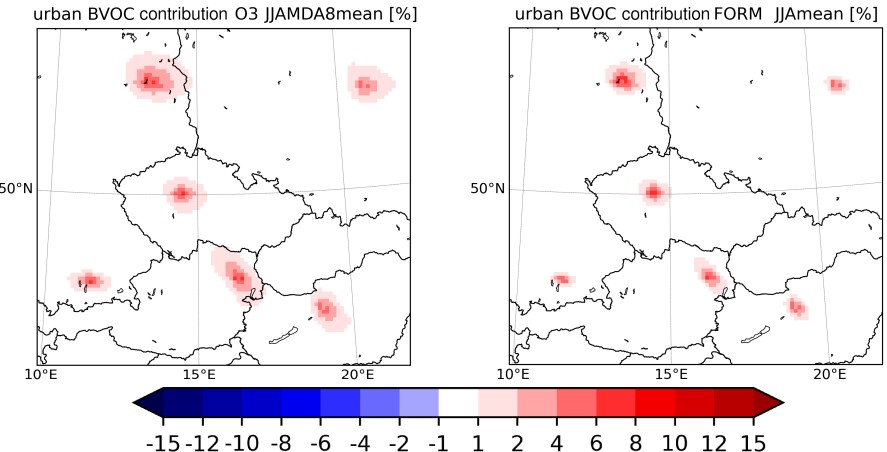

**Figure 18.** The relative contribution of the impact of urban BVOC emissions to the total impact in % for the MDA8 ozone (left) and daily mean HCHO (right) averaged over 2007-2016 JJA. The plot is zoomed to the cities analyzed.

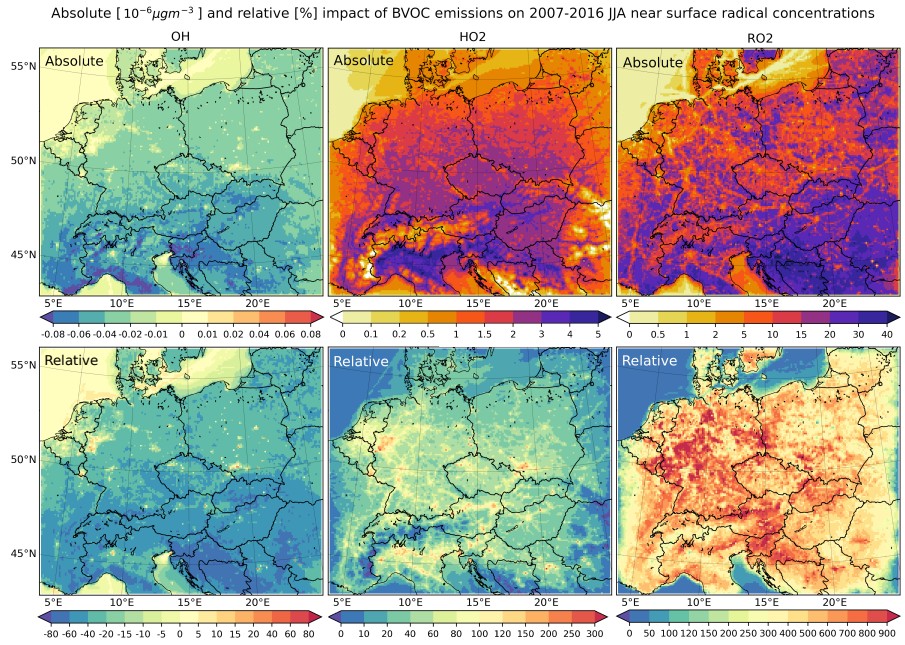

**Figure 19.** The absolute (top) and relative (bottom) impact of all BVOC emissions on the near surface concentrations of OH (left), $HO_2$ (middle) and $RO_2$ (right) averaged over 2007-2016 JJA. Units are $pgm^{-3}$ ($10^{-6} \mu gm^{-3}$) or % for the relative impact.

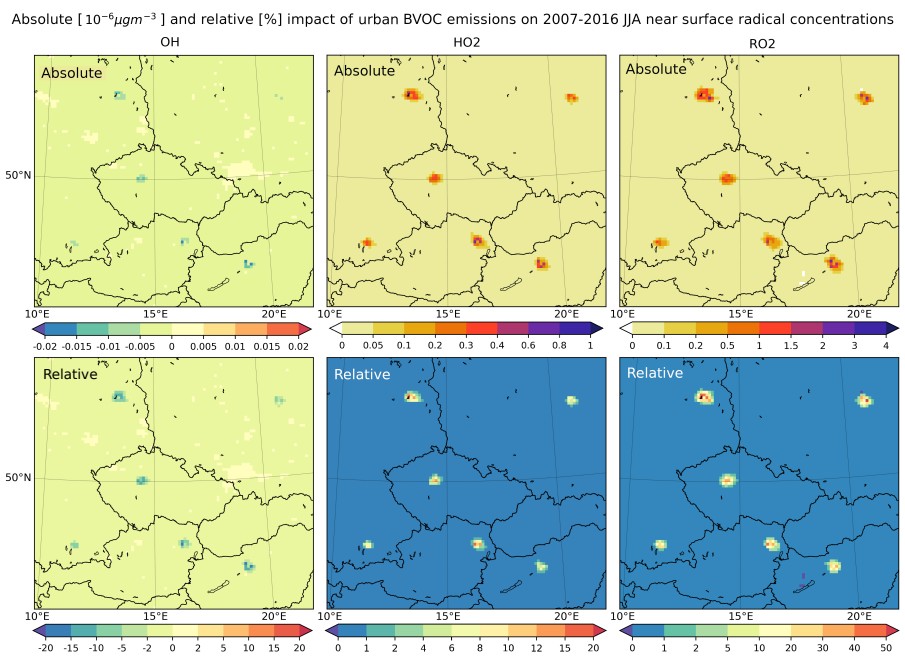

**Figure 20.** The absolute (top) and relative (bottom) impact of urban BVOC emissions on the near surface concentrations of OH (left), $HO_2$ (middle) and $RO_2$ (right) averaged over 2007-2016 JJA. Units are in $pgm^{-3}$ ($10^{-6}\mu gm^{-3}$) or % for the relative impact. The figure is zoomed to the cities analyzed.

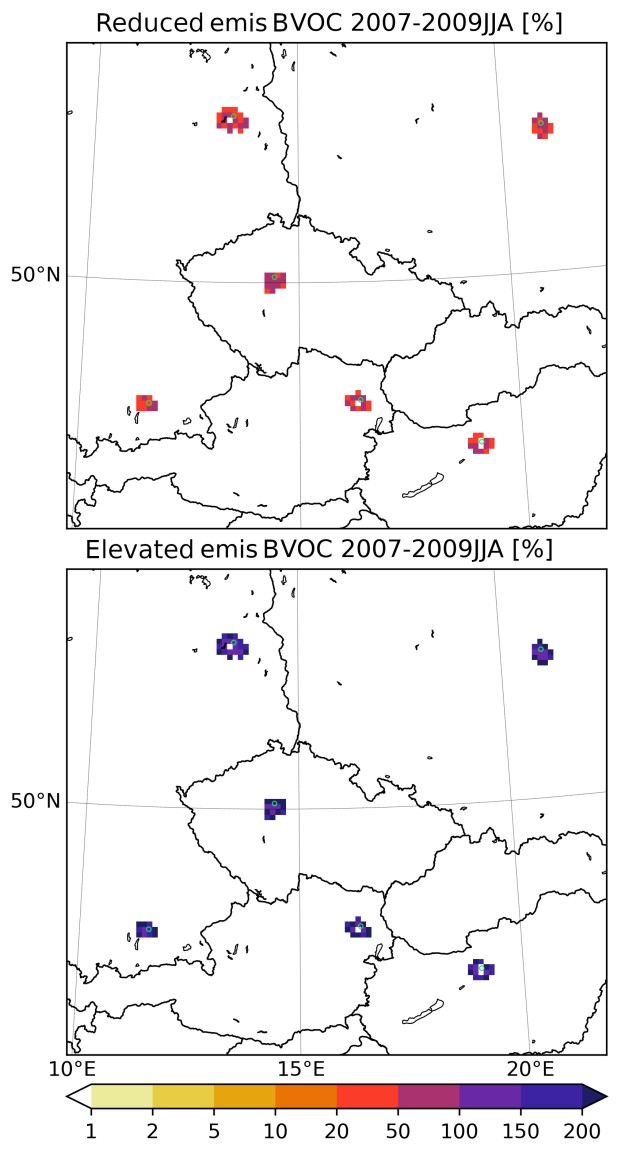

**Figure 21.** The urban BVOC emission used in the 0.5nuBVOC (top) and 2nuBVOC (bottom) experiments relative to the default urban emission fluxes seen in Fig. 1. Units in %.

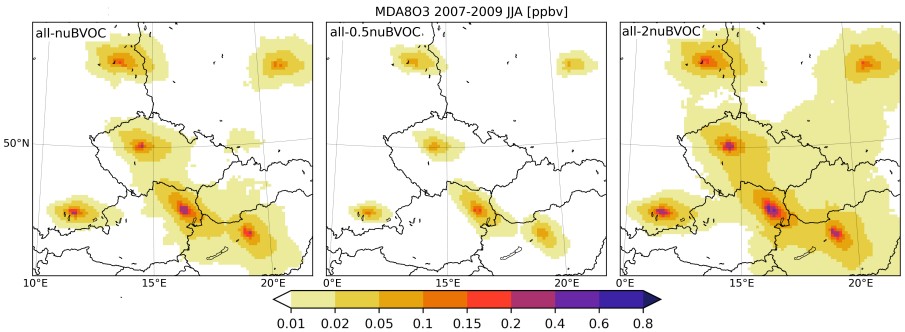

**Figure 22.** The sensitivity of MDA8 ozone on the amount of urban BVOC emissions in ppbv for the default situation (allBVOC-nuBVOC; left), and the modified urban fraction of the BVOC emissions (allBVOC-0.5nuBVOC; middle and allBVOC-2nuBVOC; right) averaged over 2007-2009 JJA. The plot is zoomed to the cities analyzed.

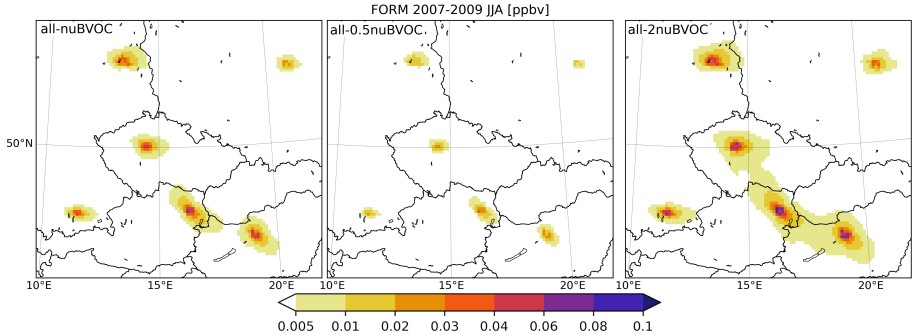

**Figure 23.** Same as Fig. 22 but for formaldehyde.

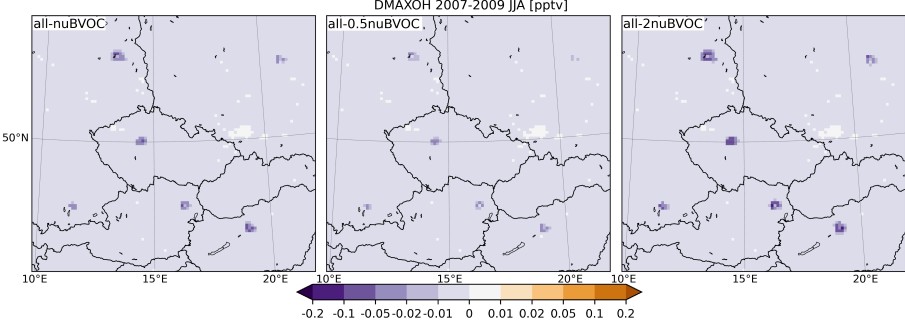

**Figure 24.** Same as Fig. 22 but for OH and in $\mathrm{pgm}^{-3}$ ($10^{-6}\mu gm^{-3}$).