# Peer review of "The long-term impact of BVOC emissions on urban ozone patterns over central Europe: contributions from urban and rural vegetation"

_EGUsphere, 2024_

## Referee Comment (RC2)

**Review of "The long-term impact of BVOC emissions on urban ozone patterns over central Europe: contributions from urban and rural vegetation"**
Manuscript ID: egusphere-2024-2027

General comments:

The authors explored the impact of urban BVOC emissions on atmospheric oxidants, including O3, HCHO and OH, over a decade (2007-2016) in central Europe by using the MEGAN model and WRF-CMAx. However, in the results and discussions, the authors have discussed the impact of BVOCs emissions in the past decade by averaging them, which ignores the annual changes in BVOCs emissions caused by variation in meteorology, land type, and vegetation during the year 2007-2016, and the impact of these changes on the concentrations of atmospheric oxidants. This is also inconsistent with the "long-term impact of BVOC emissions" which proposed in the manuscript title. Long-term changes in BVOCs emissions and their impact on atmospheric oxidant concentrations over decadal periods should be of interest.

In addition, the MEGAN model used in this study to calculate BVOCs emissions should not only consider the impact of the changes in meteorological fields which provided by the WRF model, but also consider the changes in land type, LAI, and vegetation type. This may lead to uncertainty in the estimated BVOCs emissions, and thus affect the estimation of its contribution to atmospheric oxidants concentrations. The authors should discuss the uncertainties in the BVOCs estimated by MEGAN model and the WRF input data, and the impact of these uncertainties on evaluating the impact of urban BVOCs on ozone.

The figures in the current manuscript should be further integrated and optimized. The discussion and conclusion section should present the discussion and outlook of the current research work, rather than repeating the results of the manuscript. This section seems too long, should further summarize the findings and conclusions.

Overall, the research content of this manuscript is quite interesting and is currently a hotspot in the field. However, the writing and figures need improvement to meet the ACP journal's standards. The current version of this manuscript requires major revisions before it can be considered for publication.

Specific comments:

1. Line 72: The first time an OSAT appears, its full name should be provided.
2. Line 79-80: It is mentioned here that the interplay of anthropogenic and biogenic VOC emissions is synergic. How anthropogenic VOC emissions and the interplay between them were considered in setting up model experiments in this study?
3. Line 82-83: "The dominant role of natural VOC emissions over anthropogenic ones", what does this mean?
4. Line 81-93: The literature listed here seems messy and illogical. We suggest that the author need to further improve the introduction section. Also, there are studies on the impact of BVOC emissions on air quality in urban in China, such as Ma et al. (2021). Authors should consider when conducting literature research.
   Reference: Ma, M., Gao, Y., Ding, A., Su, H., Liao, H., Wang, S., ... & Gao, H. (2021). Development and assessment of a high-resolution biogenic emission

inventory from urban green spaces in China. Environmental science & technology, 56(1), 175-184.

5. Line 194-207: For MEGAN model, what are the specific land cover types used in the model? What is the data source and the base year of land cover types? The authors focus on ten years (2007-2016). Does the land cover type change during this decade? Does the MEGAN model consider the impact of changes in land type on BVOC emissions? If there is a difference between the base year of land cover types and the study year, will this difference affect the calculation of BVOC emissions?

What are the criteria for matching land cover types with vegetation types in MEGAN?

Are the soil temperature and soil moisture provided by the WRF simulated results? Are there any biases between the soil temperature and moisture simulated by the model and observations? How much uncertainty will these biases lead to the simulation of BVOC emissions? For $CO_2$ concentration in MEGAN model, does it a fixed value or something else?

6. Line 254: Should use BVOC or biogenic VOC? The author needs to unify.

7. Line 255-266: The authors compared the BVOC emissions calculated by MEGAN and CAMS-GLOB-BIO. What are the differences in the parameterization schemes for calculating BVOC emissions? If the differences are only due to land cover type and meteorological fields, the authors should provide more detailed explanations on how the differences in meteorological fields affect the simulated BVOC emissions.

8. Line 278-280: Does "2nuBVOC" and "0.5nuBVOC" mean changing the fraction of BVOC emissions in urban areas within the grid? How are BVOC emissions in urban and nonurban areas defined in this study?

9. Line 305: Need to mark Figure 5 in this paragraph. Also, the title of Figure 5 should indicate that it is the average over the 10 years (2007-2016).

10. Line 317: Does the 2-5 here mean 2-5%?

11. Line 318-325: For Figure 6, how does the impact of BVOC emissions on ozone change between different seasons from the year 2007 to 2016? It is recommended that the author provide the average annual changes in the impacts during different seasons in year 2007-2016.

12. Line 326-353: The impact of BVOC on ozone, formaldehyde and OH over city surrounding and urban centers are both kind of different, which can be further explained based on the differences in BVOC emissions and distribution.

13. Line 355-368: The author plot both Figure 10 and Figure 11. I can understand that Figure 10 shows the contribution of urban BVOCs to O3, HCHO and OH concentrations, while Figure 11 shows the impact of urban BVOCs on these pollutants. However, the author did not figure out why Figures 10 and 11 have different distribution of contributions and impact on O3. Also, there seems to be no difference between these two calculation methods for HCHO and OH.

14. For Figure 12-14, suggest author recompose these figures. The current figures are difficult to understand the impact of urban BVOCs emissions between city centres

and city vicinities.

15. Line 384-391: What does the meaning of "relative share".

16. It is difficult to tell from the colorbar in Figure 15-17 whether it is a positive or negative contribution or impact of urban BVOCs. The author can represent positive contributions with warm colors and negative contributions with cool colors.

17. Line 421-422: According to Figure 18, the legends are all positive values. How can you conclude that the urban BVOC emissions have decreased by 50% compared to the default case?

18. Line 527: Is the difference in urban BVOC emissions between the two calculations just a difference from BVOC emissions?

---

## Author Comment (AC1)

**Authors' responses to referee comments on egusphere-2024-2027 titled "The long-term impact of BVOC emissions on urban ozone patterns over central Europe: contributions from urban and rural vegetation"**

Referee comments 1:

Dear Anonymous Referee #1,

thank you for your time and effort to review our paper and for all your comments. Please find our point-by-point answers to the points of your revision (in bold italic) below.

Liaskoni et al. study showed long-term (2007-2016) impact of Biogenic Volatile Organic Compounds (BVOC) emissions on urban ozone as well as formaldehyde and OH over central Europe. The study also explored the partial role of the urban vegetation and evaluated its share in the overall ozone formation due to all BVOC emissions. The study further assessed the changes in the oxidative capacity of the atmosphere by considering the oxidants e.g. OH and peroxy radicals and the dominant oxidation product of BVOCs (formaldehyde). Finally he study conducted a couple of sensitivity analyses to assess the uncertainty arising from the calculation of the urban fraction of BVOC emissions. This is an interesting paper as ozone can be considered as a regional problem, and the impact of vegetation emissions on urban areas is still unclear to some extent. Thus, the long-term regional impact of BVOC over decadal times-scales over central Europe can have а substantial contribution to scientific progress within the scope of this journal. The scientific approach and methods used in the study is perfect, but the paper is written in extremely bad English with grammatical errors and simple wording which doesn't go into the detail and causes confusion of the reader. Therefore, the article needs a major revision in its current form, after which it can be reviewed again for publication consideration.

Authors response: Thank you for the positive evaluation of the scientific approach and the methods applied in our paper and we completely agree that the English language needs substantial improvements throughout the manuscript. We therefore included all the corrections related to language you are mentioning in your review. We further revised the entire text with English language expert. We hope the revision meets the acceptable language standards for scientific papers.

The model validation results showed the overestimation of ozone in urban areas and even in rural areas. The explanation for the bias has been discussed with appropriate references. Can the authors include NOx model-measurement comparison to validate the model? NOx will play the major role for ozone variation in urban areas. This results can give more idea about the deviation of ozone in urban areas. It is also important to include how the model-measurement bias can impact on the overall results.

Authors response: we agree that given the crucial role of NOx in ozone formation, especially in over NOx-rich environments as city centres, the model validation for NOx should be also included in the manuscript. Therefor we added two figures in the manuscript representing the annual and diurnal cycles of the measured and modelled NO2 concentrations of both rural and urban stations while we chose all stations that were used for ozone which measured also NO2. We extended accordingly the text in the Validation section. We also extended the Discussion to support the explanation of ozone biases by the encountered NO2 biases.

**The details of the measurement data is absent in Method section. And also how the cities have been selected (on what parameters: NOx level or vegetation coverage)?**

Authors response: We included additional information about the stations chosen for the validation: all stations are chosen from the AirBase air quality database while for ozone and for the rural stations, 117 stations (from Austria, Czechia, Germany, Hungary, Poland and Slovakia i.e. countries that cover most of the computational domain) are selected that cover the modelled time period. For NO2, we chosen those ozone measuring stations which measure also NO2. Only stations up to 800 m ASL are selected due to model resolution that cannot represent high elevation orography. For the chosen cities, all urban and suburban background stations were selected regardless of the NOx level or vegetation cover around the station. We added supplemental material to the manuscript with the exact list of station codes (using the AirBase nomenclature) used in the study.

**Has the shading effect included in the MEGAN model? If not, how this can have impact on the overall emissions of BVOCs.**

Authors response: Building shading can indeed play an important role in modulating the incoming solar radiation on vegetation surfaces, especially near the northern faces of buildings, however this is not accounted for in the MEGAN model and in fact most of the vegetation represented in the datasets correspond to open areas like parks, urban forests, leisure areas where the vegetation is not so affected by building shades. However, we can assume that disregarding this effect might introduce some degree of overestimation of the BVOC flux. On the other hand, this is impossible to properly account for at the chosen resolution as this would need very detailed landcover data reaching street levels in which the vegetation might be significantly affected by shading. This was noted in the revised manuscript. We also referred to a recent study of Maison et al. (2024, ACP) who used relatively high resolution of 1 km compared to our 9 km for Paris, but even at this resolution they did not account for shading effects.

**Some other comments:**

**Line 9: Most of the domain. What domain? Need to clarify.**

Authors response:

We replaced the "domain" (which referred to the computational domain/grid) by the "modelled region", which is more informative,

**Line 25-26: Will they be monoterpenes, sesquiterpenes?**

Authors response:

Yes, this is right. These are families of compounds, so plural is needed here.

**Line 38-39: This statement should be applicable at high NOx condition which needs to be mentioned.**

Authors response: Yes, this requires to have sufficient NOx for the reaction of NO with peroxides and for recycling OH. We added this to the revised text.

**Line 39: a comma (,) or first bracket () need to be added after organic peroxy radicals.**

Authors response: Bracket added.

**Line 43: 'nitrate radical (NO3)' has already been defined before, so you can remove 'nitrate radical' from here.**

Authors response: deleted.

**Line 49-50: This has already been written in Line 38-39. You can delete anyone.**

Authors response: Merged the information from this paragraph and the one before to avoid repeated information/statements.

**Line 83: (Richards et al., 2013) will be Richards et al. (2013).**

Authors response: Corrected.

**Line 93: 'to' needs to be added after 'due'.**

Authors response: Added.

Line 109: 'Nowak et al. (2000), analysing various micro-climatic conditions above the urban domain of Washington DC to Massachussets, showed that' can be written as 'Nowak et al. (2000) showed by analysing various micro-climatic conditions above the urban domain of Washington DC to Massachussets that'

Authors response: Corrected.

**Line 143: To used BVOC emissions model?- Please correct the sentence.**

Authors response: Corrected to "The BVOC emission model used as well as ...".

**Line 155: 'as well' will be 'as well as'.**

Authors response: Corrected.

**Line 179: typo 'inbtermediates'**

Authors response: Corrected.

**Line 170: which monoterpene as a representative of monoterpenes is used in the mechanism? There is a large variation of rate coefficients and products for different monoterpenes oxidation process. How can this impact on the overall results?**

Authors response: In the used chemical mechanism (CB6r5), monoterpenes are represented by one lumped group named TERP and, as already mentioned in the manuscript, their oxidation with OH, O3 or NO3 is represented by one summary reaction (one for each oxidant). The reaction rates for these summary reactions are based on the older chemical mechanism CB05 (Sarwar et al, 2008) in which the TERP oxidation follows that in the SAPRC99 (Carter, 2000) mechanism. In SAPRC99, the reaction rates are averaged based on chamber measurement of the oxidation of an average mixture of monoterpenes. We agree that such averaging brings some errors to the results due to non-linearity of the chemical reactions, but due to the long simulations some compromise is necessary between numerical feasibility and accuracy and explicitness of the chemical mechanism. This is noted in the "Chemical transport model" section where the chemical mechanism used (including the way BVOC oxidation is represented) is described.

Ref:

Carter, W. P. L.: DOCUMENTATION OF THE SAPRC-99 CHEMICAL MECHANISM FOR VOC REACTIVITY ASSESSMENT, Final Report to California Air Resources Board, Contract 92-329 and Contract 95-308, 00-AP-RT17-001-FR, University of California, Riverside, California 92521, 2000.

Sarwar, G., Luecken, D., Yarwood, G. Whitten, G. Z. and Carter, W. P. L.: Impact of an Updated Carbon Bond Mechanism on Predictions from the CMAQ Modeling System: Preliminary Assessment, J. App. Meteorol. Clim., 47, 3-14, https://doi.org/10.1175/2007JAMC1393.1, 2008.

**Line 182: HOx recycling and its impact on NOx-ozone chemistry (Archibald et al., 2010) has been updated in Khan et al. (2021). You could include this information in here.**

**Ref: Khan et al. (2021) Changes to simulated global atmospheric composition resulting from recent revisions to isoprene oxidation chemistry. Atmospheric Environment 244, 117914.**

Authors response: Thank you for informing us about the paper with updates. We included this information in the revised manuscript.

**Line 197: 'sesquiterpene' will be 'sesquiterpenes'.**

Authors response: Corrected.

**Line 234: Please define 'JJA' when you use it first time.**

Authors response: Defined.

**Line 236: 'of' needs to be added after role.**

Authors response: added.

**Line 249: 'on' needs to be added after depending.**

Authors response: added.

**Line 258: 'some' needs to be removed.**

Authors response: Removed and the sentence was a bit modified.

**Line 260: the has been written two times.**

Authors response: The second occurrence removed.

**Line 297: 'then' will be 'than'**

Authors response: Corrected.

**Line 301: 'here' should be replaced by the figure number 'Figure 5 to Figure 9'.**

Authors response: We rather replaced "here" to "in this section" and kept the reference for individual figures in the text where the results are commented for the particular figure.

**Line 305: Need to mention the figure number 5 in this sentence.**

Authors response: Figure reference added.

**Line 307: 'norther' will be 'northern'**

Authors response: Corrected.

**Line 317: The sentence is not complete.**

Authors response: The sentence was corrected to "Over areas without BVOC emissions (sea) however OH increases by around 0.01-0.02 pptv (2-5%)."

**Line 318: 'ozone' needs to be added after MDA8.**

Authors response: Added.

**Line 319: You could include the names of the urban areas in here.**

Authors response: The six name where added (i.e. Berlin, Budapest, Munich, Prague, Vienna and Warsaw)

**Line 321: 'that' has been written two times.**

Authors response: The second occurrence removed.

**Line 347: 'well know' will be 'well known'.**

Authors response: Corrected.

**Line 382: 'reaches' has been written two times.**

Authors response: Corrected.

**Line 384: 'as' will be 'us'.**

Authors response: Corrected.

**Line 392: Impact on hydrogen oxide radicals. Why RO2 has been added in HOx?**

Authors response: Yes, this was a mistake, we corrected the title of the subsection to "Impact on hydroxyl and peroxide radicals (OH, HO2, RO2)"

**Line 410: 'to' needs to be added after up.**

Authors response: Added.

**Line 426: unit missing after 0.02-0.04.**

Authors response: Units added (ppbv).

**Line 427: The sentence 'Thus, again, half and twice of the default case, respectively' does not make any sense.**

Authors response: Corrected to "These numbers represent, again, half and twice the ones in the default case, respectively."

**Line 437: Is the sentence 'overestimation over rural areas up to 20 $\mu$ gm-3 while up to 30 $\mu$ gm-3)' correct?**

Authors response: We meant "...while up to 30  $\mu$ gm-3 over cities". "Cities" was added.

**Line 440: The sentence has grammatical error.**

Authors response: The sentence was corrected to "It is probably caused mainly by the large night-time overestimation of ozone while the daytime values are captured more accurately"

**Line 443: Im et al. (2015) needs to be changed with (Im et al., 2015).**

Authors response: Corrected.

**Line 443: 'out' will be 'our'.**

Authors response: Corrected.

**Line 449: The sentence with reference Zhu et al. (2024) is not clear to me.**

Authors response: The sentence points out the fact that when the chemical transport model is applied in coarse resolution that the concentrated city center NOx emissions are not resolved and they are instead diluted to the model grid so the local increase of NOx concentrations is not high enough to the first order ozone titration – instead they efficiently cause ozone production. This behavior was recently seen in Zhu et al.(2024) where the daily ozone maxima were overestimated for the mentioned reason. This is clarified in the manuscript.

**Line 457: Grammatical error. Need to add 'have' after We**

Authors response: Added.

**Line 474: The sentence is not correct. There or Therefore?**

Authors response: The sentence starts with There to refer to city centers. However, this is not very comprehensible, so we changed it to: Over them, the BVOC emissions are very small while the ozone increases were large causing OH increases due to direct production from atomic oxygen.

**Line 481: chemical can be written as 'species'.**

Authors response: Replaced.

**Line 484: 'os' will be 'is'**

Authors response: Replaced.

**Line 512: causes will be cause**

Authors response: Replaced.

**Line 520: 'Crigee' will be 'Criegee'**

Authors response: Corrected.

**Line 521: 'pof' will be 'of'**

Authors response: Corrected.

**Line 532: unit missing after 0.2 to 0.8**

Authors response: ppbv added.

**Figure 17: 'in' should be included after 'Unites are'**

Authors response: Included.

**Figure 16 and Figure 5 OH distribution should be similar? If they are similar, why they are**

**shown in two places?**

Authors response: These are not completely the same distributions. In Fig 5, the average daily maximum OH is presented, which is a representative measure of the daytime oxidizing capacity during maximum solar insolation. On the other hand ,Fig 16 shows the average OH (averaged trough all hours for JJA 2007-2016). The decision to include OH again was that we wanted to show the impact on all relevant radicals, i.e. HO2, higher RO2 but also OH, so we decided to show OH again, albeit as daily mean (to be consistent with the HO2 and RO2 figure.)

**Peroxydes and hydroperxydes need to be corrected by Peroxides and hydroperoxides throughout the manuscript.**

Authors response: Corrected at all occurrences.

Referee comments 2:

Dear Anonymous Referee #2,

Thank you for your time and effort to review our paper and for all your detailed comments and criticism to guide us to improve the manuscript . Please find our point-by-point answers to the points of your revision (in bold italic) below.

**General comment:**

The authors explored the impact of urban BVOC emissions on atmospheric oxidants, i ncluding O3, HCHO and OH, over a decade (2007-2016) in central Europe by using the MEGAN model and WRF-CMAx. However, in the results and discussions, the authors have discussed the impact of BVOCs emissions in the past decade by averaging them, which ignores the annual changes in BVOCs emissions caused by variation in meteorology, land type, and vegetation during the year 2007-2016, and the impact of these changes on the concentrations of atmospheric oxidants. This is also inconsistent with the "long-term impact of BVOC emissions" which proposed in the manuscript title. Long-term changes in BVOCs emissions and their impact on atmospheric oxidant concentrations over decadal periods should be of interest. In addition, the MEGAN model used in this study to calculate BVOCs emissions should not only consider the impact of the changes in meteorological fields which provided by the WRF model, but also consider the changes in land type, LAI, and vegetation type. This may lead to uncertainty in the estimated BVOCs emissions, and thus affect the estimation of its contribution to atmospheric oxidants concentrations. The authors should discuss the uncertainties in the BVOCs estimated by MEGAN model and the WRF input data, and the impact of these uncertainties on evaluating the impact of urban BVOCs on ozone.

The figures in the current manuscript should be further integrated and optimized. The discussion and conclusion section should present the discussion and outlook of the current research work, rather than repeating the results of the manuscript. This section seems too long, should further summarize the findings and conclusions. Overall, the research content of this manuscript is quite interesting and is currently a hotspot in the field. However, the writing and figures need improvement to meet the ACP journal's

**standards. The current version of this manuscript requires major revisions before it can be considered for publication.**

Authors response: Thank you for your detailed comment. We admit that the interannual variability of MEGAN emissions requires more attention and as well as the interannual variation of its inputs, i.e. LAI, PFT and meteorological input. We therefore included in the paper a more detailed and clear description of what input is a 10yr average and what changes trough time with justification for our choice (including some support figure for LAI). See for details below in the specific comments.

As for the figures, we made substantial improvements, mainly for the color bars and ranges chosen (e.g. if they present both negative and positive values, we chose a centered colorbar with cold/warm colors for negative/positive values etc.).

We made also modifications in the Discussion section, which is although long, but we followed the practice that the Results section presents the results only with minimum interpretation and discussion while it is the Discussion section which carefully goes through all the results and put them into context with adequate explanation/interpretation (including comparison with past studies/results).

We also added a supplement to our manuscript with a 1) list the stations used in the validation (as aked by the other referee) and 2) to show the interannual variability of the BVOC impact to address the referee's comments/criticism raised below.

**Specific comments:**

**1. Line 72: The first time an OSAT appears, its full name should be provided.**

Authors response: We included the full name, i.e. Ozone Source Apportionment Technology, in the revised manuscript.

**2. Line 79-80: It is mentioned here that the interplay of anthropogenic and biogenic VOC emissions is synergic. How anthropogenic VOC emissions and the interplay between them were considered in setting up model experiments in this study?**

Authors response: The synergical co-acting of anthropogenic and biogenic emissions is inherent of the simultaneous emission of both family of VOCs and their consequent chemical evolution leading the changes in ozone, FORM and radical concentrations. However, the study's goal was mainly to evaluate the partial role of the biogenic fraction of VOC emissions therefor we chosen the so called zero-out (annihilation) method by setting up an experiment where BVOC where not considered at all (only anthropogenic emissions). We could have set up a further simulation where all VOC emissions had been zeroed out and calculated the impact of anthropogenic emission only (and then finding potentially out that the sum of the separate effect of anthropogenic emissions and biogenic emissions is lower than the effect of both acting together) but this this would mean a drift from the original goal of the study dealing with the impact of BVOC emissions only.

**3. Line 82-83: "The dominant role of natural VOC emissions over anthropogenic ones", what does this mean?**

Authors response: This means that the simulated ozone concentrations are mostly sensitive to natural VOC emissions (i.e. BVOC), as the cited paper (Richards et al., 2013) states: "Our results show a dominant sensitivity to natural VOC emissions in the Mediterranean basin over anthropogenic VOC emissions". We added a note on this in the revised manuscript.

**4. Line 81-93: The literature listed here seems messy and illogical. We suggest that the author need to further improve the introduction section. Also, there are studies on the impact of BVOC emissions on air quality in urban in China, such as Ma et al. (2021). Authors should consider when conducting literature research.**

**Reference: Ma, M., Gao, Y., Ding, A., Su, H., Liao, H., Wang, S., ... & Gao, H. (2021). Development and assessment of a high-resolution biogenic emission inventory from urban green spaces in China. Environmental science & technology, 56(1), 175-184.**

Authors response: We modified the introduction section a bit, specifically the part from Line 38 (see the track changes version of the revised manuscript). We added new paragraphs describing some of the modelling studies to reveal the impact of BVOC on urban ozone focusing on eastern China and also US and Canada and also included some new citations into the following part which summarizes the literature regarding the role of the urban vegetation (including the reference suggested by the reviewer, which turned to be very useful and had been omitted by us before).

**5. Line 194-207: For MEGAN model, what are the specific land cover types used in the model? What is the data source and the base year of land cover types? The authors focus on ten years (2007-2016).**

Does the land cover type change during this decade? Does the MEGAN model consider the impact of changes in land type on BVOC emissions? If there is a difference between the base year of land cover types and the study year, will this difference affect the calculation of BVOC emissions?

What are the criteria for matching land cover types with vegetation types in MEGAN? Are the soil temperature and soil moisture provided by the WRF simulated results?

**Are there any biases between the soil temperature and moisture simulated by the model and observations? How much uncertainty will these biases lead to the simulation of BVOC emissions? For CO2 concentration in MEGAN model, does it a fixed value or something else?**

Authors response: For the calculation of BVOC emissions, MEGAN distinguishes 16 so called plant-functional-types (PFT) which can potentially emit BVOCs. These are groups of plants that have similar functional behavior (e.g. Needleleaf Evergreen Temperate Tree, Needleleaf Deciduous Boreal Tree, Broadleaf Evergreen Tropical Tree, Arctic Grass, Cool Grass, Crop and so on). These are provided as fractions of the gridcell and are based on MODIS satellite observation based on the approach of Lawrence and Chase (2007). The PFT data used in our study is for 2010. Another important input to module BVOC emissions from these PFTs is the leaf-area-index (LAI) which accounts for the annual cycle of the total area of plant leafs (with a minimum during winter and maximum in summer, ranging from 0 to 8). LAI is also based on MODIS and is from the same year 2010. It is clear that in general, both inputs vary between years but we made the assumption that this year-by-year variation is negligible during the period studied (2007-2016).

To justify this, for LAI we calculated to domain mean monthly values for the entire period using the data for each year and the results are found in a new Figure 3 (in the revised manuscript). It clearly shows that the LAI during summer (which is of interest for us as we examined only the summer impact) is almost the same for each year and varies between about 2.6-2.7.

As for the evolution of the landuse, more specifically the evolution of PFTs, we rely on the fact that during the studied period the total amount of farmland (made mainly of crops) changed negligibly, see the Eurostat report on the farm land evolution between years 2005 to 2020 (https://ec.europa.eu/eurostat/statistics-explained/SEPDF/cache/73319.pdf)

Another large source of BVOC over the studied domain are forests but European forest are well protected and the trends in forests area are negligible (less than 1% over central Europe, i.e. the study area), see https://foresteurope.org/wp-content/uploads/2016/08/SoEF\_2020.pdf.

In other words, the landuse types responsible for emitting the majority of BVOC have not changed considerably during the studied period so taking one year as representative is thus justified. Of course, the vegetation in urban areas has changed during this time as urban development is an ongoing process. This is thus the only uncertainty introduced in the study, and we admit that more detailed description of the urban vegetation including its evolution must be accounted for in future ozone/urban-related studies.

In the revised manuscript, we included the above-mentioned facts and sources in the section describing the input data that was used to drive MEGAN.

As for the soil temperature and soil moisture, both are input to the MEGAN model. Soil temperature is however only used to calculate soil NOx emissions (which we were not focused on in this study). The soil moisture from WRF was used in MEGAN for isoprene emissions for the correction factor accounting for "wilting point" (i.e. soil moisture below which plants cannot extract water from the soil resulting in zero emissions) (Guenther et al, 2012). We did not perform a comparison of WRF soil moisture with observation but rely on earlier studies that made such evaluations, e.g. the soil moisture comparison with station observations in Italy showed that Noah shows the best performance (Zhuo et al., 2019). In our study, we also used the Noah landsurface scheme so we can expect that it performed well also in our simulations regarding the estimation of moisture in soil.

Finally, the CO2 dependence of BVOC emissions is included in the MEGAN model with uniformly distributed CO2 in space, however, annual temporal evolution is accounted for.

Literature:

Lawrence, P. J. and Chase, T. N.: Representing a new MODIS consistent land surface in the Community Land Model (CLM 3.0), J. Geophys. Res.-Biogeo., 112, G01023, https://doi.org/10.1029/2006JG000168, 2007.

EUROSTAT, 2020:

https://ec.europa.eu/eurostat/statistics-explained/SEPDF/cache/73319.pdf (crop land evolution 2005 vs 2020.)

FORESTEUROPE, 2020:

https://foresteurope.org/wp-content/uploads/2016/08/SoEF\_2020.pdf (change in forest area between the 90s and 2020)

**6. Line 254: Should use BVOC or biogenic VOC? The author needs to unify.**

Authors response: We unified the use BVOC across the manuscript instead of biogenic BVOC.

**7. Line 255-266: The authors compared the BVOC emissions calculated by MEGAN and CAMS-GLOB-BIO. What are the differences in the parameterization schemes for calculating BVOC emissions? If the differences are only due to land cover type and meteorological fields, the authors should provide more detailed explanations on how the differences in meteorological fields affect the simulated BVOC emissions.**

Authors response: The reason for these differences is probably in the fact that the CAMS-GLOB-BIO 3.1 version of these data incorporated an updated region-specific emission factors (EF) for different plants instead of using the default MEGAN emission factors (used in version CAMS-GLOB-BIO2.1 as well as in our study) and this resulted in lower emissions over Europe compared to the default EF used in our set-up (see the difference of BVOC emissions between version 2.1 and 3.1 in the mentioned study, Sindelarova et al.,2022).

Some difference can be accounted for the difference between the meteorology in WRF vs. the one used in CAMS-GLOB-BIO3.1. WRF was driven by the older ERA-Interim data so we might expect that the WRF generated near-surface fields are close to the reanalysis values and are thus slightly overestimated. Indeed, Karlicky et al.(2018) who used WRF driven by ERA-Interim over the same domain/resolution and with a similar set of parameterizations showed also some overestimation of near surface temperatures in central Europe.

On the other hand, CAMS-GLOB-BIO3.1 was driven by ERA5 and it was shown by many that ERA5 has slightly lower temperatures (which reduced the

higher ERA-Interim bias). This means that BVOC emission fluxes in CAMS-GLOB-BIO3.1 could be lower due to this reason too.

We made this clear in the revised manuscript.

Reference:

Karlický, J., Huszár, P., Halenka, T., Belda, M., Žák, M., Pišoft, P., and Mikšovský, J.: Multi-model comparison of urban heat island modelling approaches, Atmos. Chem. Phys., 18, 10655–10674, https://doi.org/10.5194/acp-18-10655-2018, 2018.

**8. Line 278-280: Does "2nuBVOC" and "0.5nuBVOC" mean changing the fraction of BVOC emissions in urban areas within the grid? How are BVOC emissions in urban and nonurban areas defined in this study?**

Authors response: The urban fraction was calculated by masking out the PFT data by the city boundaries in a way that first it was calculated that how much of the gridcell falls within the urban area and then this factor was applied to the PFT fractional data allowing to calculate the emissions of BVOC from this fraction. We admit that this information was not clearly stated in the manuscript.

Regarding 2nuBVOC and 0.5nuBVOC they mean that the urban fraction of the original PFT grid fraction were doubled and halved (to account for the sensitivity to the urban fraction of BVOC emissions given the simple estimation of this fraction by taking "what falls within the city" approach).

**9. Line 305: Need to mark Figure 5 in this paragraph. Also, the title of Figure 5 should indicate that it is the average over the 10 years (2007-2016).**

Authors response: The figure has been marked and we added the 2007-2016 to the figure caption.

**10. Line 317: Does the 2-5 here mean 2-5%?**

Authors response: yes, the "%" was added.

11. Line 318-325: For Figure 6, how does the impact of BVOC emissions on ozone change between different seasons from the year 2007 to 2016? It is recommended that the author provide the average annual changes in the impacts during different seasons in year 2007-2016. Authors response: As the main focus of the manuscript is summer conditions which exhibit the largest BVOC emissions and most efficient ozone production occurs during summer, we limited our presentation of the spatial and diurnal impacts on this season, however, we agree with the referee, that in order to show the year-by-year variability of the summer impact within the examined decade, it would be beneficial to present he impact in different years. We therefor calculated these impacts (on ozone, formaldehyde and OH) for each year separately and the results are included in the Supplement, S3-S4 S1-S2 (ozone, absolute/relative impact), as figures (OH, absolute/relative impact) and S5-S6 (formaldehyde, absolute/relative impact).

**12. Line 326-353: The impact of BVOC on ozone, formaldehyde and OH over city surrounding and urban centers are both kind of different, which can be further explained based on the differences in BVOC emissions and distribution.**

Authors response: Indeed, the different impacts in city centres vs. surroundings is caused by i) different emissions of BVOC in cities than over surroundings (where they are higher) but also by ii) different chemical composition of the urban air compared to rural one. The ii) means that over cities, the air is rich of NOx, which causes more efficient ozone production due to the injections of BVOC which is clearly seen in the peaks of the impact on this pollutant. For formaldehyde, the impact is larger over vicinities where more BVOC is emitted and therefore more FORM is produced from oxidation of these BVOCs. Lastly, for OH, the impact is driven mainly by the oxidation of BVOC by OH reaction: over vicinities, more OH is removed via this pathway. Moreover, over cities, there is a counteracting pathway in the production of OH from ozone (via O1D). A more detailed explanation is provided in the discussion section (Lines 528 to 553 in the revised manuscript)

13. Line 355-368: The author plot both Figure 10 and Figure 11. I can understand that Figure 10 shows the contribution of urban BVOCs to O3, HCHO and OH concentrations, while Figure 11 shows the impact of urban BVOCs on these pollutants. However, the author did not figure out why Figures 10 and 11 have different distribution of contributions and impact on O3. Also, there seems to be no difference between these two calculation methods for HCHO and OH. Authors response: These figures intend to show that regarding the impact of urban BVOC, it depends to which reference state they are introduced as their impact is a function of the chemical composition of the air to which they are added.

More specifically, in the first case (Fig. 10) the urban BVOC are added to the air that already holds the rural BVOC which is clear from the calculation of the difference presented by the figure (allBVOC – nuBVOC; nu = nourban = rural). In the second case (Fig. 11), urban BVOC are added to an air which does not contain BVOC at all (noBVOC). This is a very important difference and points to a general property of the atmospheric chemistry: in general, the chemical effect of adding a chemical pollutant into the air (i.e. emitting that pollutant) depends on the state of the air to which it is added. Mathematically,

 $\Delta c_{j,i} = F(c_j, E_i)$

and not  $\Delta c_{j,i} = F(E_i)$ ,

where  $E_i$  is the emission of species i,  $\Delta c_{j,i}$  is the change of concentration of species j due to the emission of species i,  $c_j$  is the concentration of species j. I.e. the final change of concentration of species due to emission is a function (F) of the initial state to which it has been emitted.

Figures 10 and 11 intend to show the impact of this initial state (nuBVOC vs. noBVOC) and indeed, they are although similar there are still quantitative differences. For ozone, it is more evident, but also for FORM and OH it is seen that the impact (increase of formaldehyde, decrease of hydroxyl radical) is larger in case of emitting urban BVOC into a BVOC free air (i.e. noBVOC simulation being as a reference). We added some more comments on this in the Discussion part in order to explain to the reader why the impacts are larger in case of uBVOC-noBVOC difference compared to allBVOC-nuBVOC.

**14. For Figure 12-14, suggest author recompose these figures. The current figures are difficult to understand the impact of urban BVOCs emissions between city centres and city vicinities.**

Authors response: We decided to keep the colors however we changed the placement of the legend. In the revised manuscript, it is placed under the plots with clear indication which lines/colors stand for the absolute concentrations (on the left) and which lines/colors stand for the differences (BVOC impact; on the right)

**15. Line 384-391: What does the meaning of "relative share".**

Authors response: This means relative contribution. We replaced the "share" to "contribution" to be clearer in the text.

**16. It is difficult to tell from the colorbar in Figure 15-17 whether it is a positive or negative contribution or impact of urban BVOCs. The author can represent positive contributions with warm colors and negative contributions with cool colors.**

Authors response: Indeed, the colorbars/ranges for some of the figures were not chosen very well. We made several modifications almost in all 2D plots where values are both positive and negative. There we placed a symmetric colorbar with cold colors as negative values and warm ones for positive contributions.

**17. Line 421-422: According to Figure 18, the legends are all positive values. How can you conclude that the urban BVOC emissions have decreased by 50% compared to the default case?**

Authors response: This figure plots the relative change of urban BVOC emissions in the reduced (0.5nuBVOC) and elevated (2nuBVOC) case with respect to the default one, so the colors already stand for the change. In the upper case, it is (according to the colorbar) about 50% of the original emissions which means that this is a 50% decrease.

**18. Line 527: Is the difference in urban BVOC emissions between the two calculations just a difference from BVOC emissions?**

Authors response: No, the difference is made because of the difference reference state to which the urban BVOC were added. Please, refer to our answer to comment no. 13 above.